# Precise control of the interlayer twist angle in large scale MoS$_2$ homostructures

Mengzhou Liao[1,2,8], Zheng Wei[1,3,8], Luojun Du[4,8], Qinqin Wang[1,3], Jian Tang [1,3], Hua Yu[1,3], Fanfan Wu[1,3], Jiaojiao Zhao[1,3], Xiaozhi Xu [5], Bo Han[5], Kaihui Liu [5], Peng Gao [5], Tomas Polcar[2], Zhipei Sun [4,6], Dongxia Shi[1,3], Rong Yang[1,3,7✉] & Guangyu Zhang [1,3,7✉]

Twist angle between adjacent layers of two-dimensional (2D) layered materials provides an exotic degree of freedom to enable various fascinating phenomena, which opens a research direction—twistronics. To realize the practical applications of twistronics, it is of the utmost importance to control the interlayer twist angle on large scales. In this work, we report the precise control of interlayer twist angle in centimeter-scale stacked multilayer MoS$_2$ homostructures via the combination of wafer-scale highly-oriented monolayer MoS$_2$ growth techniques and a water-assisted transfer method. We confirm that the twist angle can continuously change the indirect bandgap of centimeter-scale stacked multilayer MoS$_2$ homostructures, which is indicated by the photoluminescence peak shift. Furthermore, we demonstrate that the stack structure can affect the electrical properties of MoS$_2$ homostructures, where 30° twist angle yields higher electron mobility. Our work provides a firm basis for the development of twistronics.

[1] Beijing National Laboratory for Condensed Matter Physics and Institute of Physics, Chinese Academy of Sciences, 100190 Beijing, China. [2] Faculty of Electrical Engineering, Czech Technical University in Prague, Technicka 2, 166 27 Prague 6, Czech Republic. [3] School of Physical Sciences, University of Chinese Academy of Sciences, 100190 Beijing, China. [4] Department of Electronics and Nanoengineering, Aalto University, Tietotie 3, Espoo FI-02150, Finland. [5] Electron Microscopy Laboratory and International Center for Quantum Materials, School of Physics, Peking University, 100871 Beijing, China. [6] QTF Centre of Excellence, Department of Applied Physics, Aalto University, Espoo, Finland. [7] Songshan Lake Materials Laboratory, 523808 Dongguan, Guangdong, China. [8] These authors contributed equally: Mengzhou Liao, Zheng Wei, Luojun Du. ✉email: ryang@iphy.ac.cn; gyzhang@iphy.ac.cn

Recently, two-dimensional (2D) materials and their hetero-structures have attracted a lot of attention due to their unique electrical, optical, and mechanical properties[1]. Since the weak van der Waals (vdW) interactions dominate the interlayer coupling, vdW homo- and hetero-structures can possess a degree of freedom: interlayer twist angle. Twist angle governs the crystal symmetry and can lead to a variety of interesting physical behaviors, such as Hofstadter's spectra[2,3], unconventional superconductivity[4,5], moiré excitons[6–8], tunneling conductance[9,10], nonlinear optics[11,12], and structural super-lubricity[13,14]. These initiate the age of twistronics for various electronic and photonic applications. Therefore, precise controlling the interlayer twist angle of 2D materials-based structures over a large scale is highly desired and would set a foundation for the applications of twistronics. Indeed, it is possible to fabricate the required twist angle by transfer method[4,9,15–18] or atomic force microscope (AFM) tip manipulation techniques[10,13,19,20]. However, the sample size in the previously demonstrated results is usually in the order of ten-microns, strongly impeding the applications of twistronics. Wafer-scale few-layer films were also realized[21–23], but their interlayer twist angle is random and limited by the grain size and orientation as well. To realize large-scale 2D vdW homo-/hetero-structures with accurately controlled interlayer twist angle, an approach is required, in particular for the applications of twistronics.

In this work, we report the precise control of interlayer twist angle in large-scale stacked multilayer $MoS_2$ homostructures by the combination of as-fabricated epitaxially grown oriented $MoS_2$ monolayer and water-assisted transfer technique. The interfaces of our fabricated $MoS_2$ homostructures are relatively clean since no polymer is needed to be dissolved during the transfer process. We confirm that the Raman fingerprints (low-frequency inter-layer modes and Moiré phonons), interlayer coupling, band structure, and electrical properties are strongly twist angle dependent. Considering that twisted bilayer $MoS_2$ shows a variety of fantastic physical properties, such as ultra-flatbands, shear solitons, time-reversal-invariant topological insulators, Moiré quantum well states and correlated Hubbard model physics[24–27], our work is of great significance in guiding the applications of twistronics based on large-scale 2D materials.

## Results

**Fabrication of large-scale twisted multilayer $MoS_2$ films**. In this study, wafer-scale highly oriented $MoS_2$ monolayer is fabricated by epitaxial growth technique[28] (see Methods for more details). Figure 1a illustrates a typical monolayer $MoS_2$ film on a 2-inch sapphire wafer, the optical microscope images in Supplementary Fig. 1 shows that the film is highly uniform with 100% coverage. The inset of Fig.1a is a typical low-energy electron diffraction (LEED) pattern in a random position of as-fabricated wafer-scale $MoS_2$ monolayer. Only one set of hexagonal spots with the same direction is observed, indicating that the $MoS_2$ monolayer films exhibit only 0° or 60° twin alignments with sapphire substrates[28]. Figure 1b shows the AFM image of an as-grown $MoS_2$/sapphire wafer after we scrape off the $MoS_2$ film by a tweezer. It shows that the surface of the $MoS_2$ film is clean. The thickness of the $MoS_2$ monolayer is ~0.53 nm, in good harmony with pervious results[29]. Raman and photoluminescence (PL) spectra in Fig. 1c provide further evidence that the as-fabricated monolayer $MoS_2$ film is of high quality. The line scan Raman and PL spectra of a whole wafer in Supplementary Fig. 2 also show the high uniformity of the $MoS_2$ film. For more information, please see Supplementary Notes 1.

The transfer process is illustrated in Fig. 1d. First, we use a linear guided wafer scriber to cut the whole wafer into rectangular slides (6 × 7 mm for example). As the whole $MoS_2$ film is oriented, we can use the edges of the slides to determinate the orientation

of $MoS_2$ films. Second, we use polydimethylsiloxane (PDMS) as a transfer medium and deionized water to fully peel off the $MoS_2$ films from the sapphire substrates with our home-made transfer machine. Third, we use the long or short edges to align and stamp $MoS_2$ films to target substrates layer by layer (e.g., silicon wafers with 300 nm $SiO_2$). Fourth, we directly peel off the PDMS transfer medium and the $MoS_2$ films stay intact at the target substrates. More details about the transfer process can be found in the Methods section, Supplementary Notes 1 and Supplementary Fig. 3. Figure 1e shows a series of transferred bilayer and typical trilayer $MoS_2$ homostructures with precise-controlled twist angles on the order of centimeter. The mark of stacked multilayer $MoS_2$ is based on the relative angle between adjacent transferred layers. For example, (0°,30°) trilayer indicates that the relative angle between the first and second (second and third) transferred layers is 0° (30°).

Figure 2a shows optical microscope images of three typical samples with different twist angles. The optical images indicate that the surfaces of all transferred films are clean and uniform. Moreover, the edges of each layer are very straight and sharp, ensuring the high accuracy of the twist angle. AFM images in Fig. 2b shows that the surface of the transferred monolayer $MoS_2$ film is clean and flat. For the bilayer sample, although there are some bubbles, they are less than 10% of the area, indicating the good quality of our samples. Supplementary Fig. 4a–d indicate that all the surfaces involved in the transfer process are clean and flat, no contamination was observed. Figure 2c shows the scanning transmission electron microscopy (STEM) image of the stacked bilayer $MoS_2$ with a twist angle of 30°. The Moiré pattern is close to quasicrystal bilayer graphene[30], also indicating that the interface of our $MoS_2$ film is clean. For more information, please see Supplementary Notes 3.

The electron diffraction pattern (inset of Fig. 2c) suggests that the twist angle is around 29.88°, which is very close to the designed twist angle of 30°. Notice that, due to the threefold rotation symmetry of monolayer $MoS_2$ lattice and the existence of twin lattice alignments in these films, the bilayer samples with transfer stack angles $\theta$ should have both $\theta$ and $60° – \theta$ lattice twist angles regions, which have the same electron diffraction patterns. According to previous studies, both the interlayer coupling and interlayer distance of these two structures are almost indentical[31]; the optical and electrical properties between these two structures are also similar (For example, $WSe_2/WS_2$ moiré superlattice with twist angle 0° and 60° show nearly the same triangular lattice Hubbard physics[32]). Thus, the properties of the transferred multilayer $MoS_2$ films can be controlled by the stacking angle of as-grown monolayers with twin alignments. In this paper, we directly marked all the measured twist angle within 30° for simplicity.

We also fabricated eight 30°-stacked samples and transferred them on TEM grids for further analysis. For each sample, we randomly select 10–20 positions to measure the twist angle through the electron diffraction. The statistics of the measured twist angle is shown as green bars of Fig. 2d (blue bars are mirror copy of green bars for Gaussian fitting). Based on the statistics, the distribution of twist angle is relatively narrow, and the Standard Deviation of the twist angle of our samples is $\sigma = 0.327°$. It is worth noting that, the formation of flatbands in transition metal dichalcogenide (TMD) homo- and hetero- twist structures is not so sensitive to twist angle (spanning over 1°)[33,34]. As a consequence, the accuracy of our method is enough to study TMD homo- and hetero-structures-based twistronics. For more detail discussion of the twist angle distribution, please refer to Supplementary Notes 4.

Figure 2e illustrates a typical PL spectrum of the 30° stacked $MoS_2$ bilayer, which clearly shows the peaks of B, A, and the indirect bandgap excitons. The uniformity of indirect bandgap

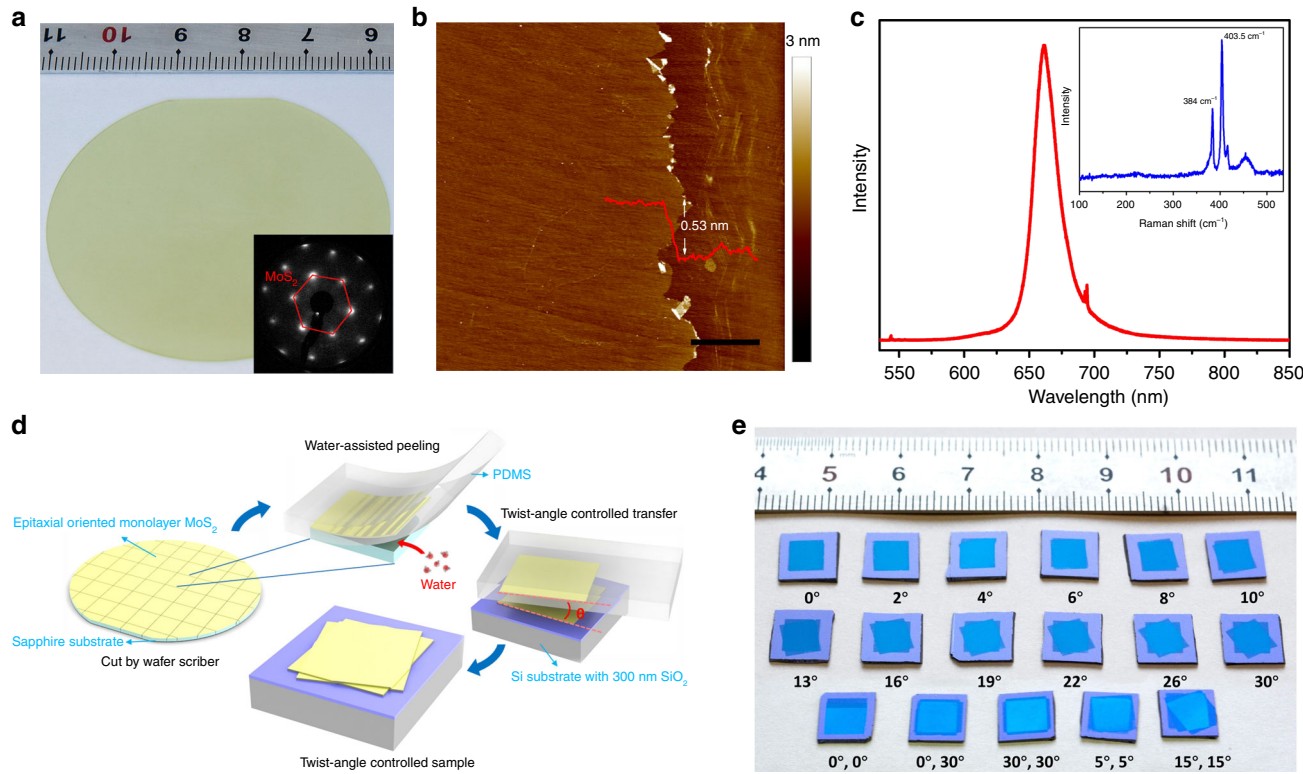

**Fig. 1 Twist angle engineering of multilayer MoS₂ homostructures. a** Image of as-grown MoS₂ monolayer on a 2-inch sapphire wafer, inset is a typical LEED pattern of the as-grown wafer. **b** AFM image of as-grown oriented MoS₂ monolayer after scraping off the right part of MoS₂ monolayer, scale bar 2 μm. The height of the film is ~0.53 nm. **c** Raman and PL spectra of as-grown MoS₂ monolayer. **d** The water-assisted transfer process. Polydimethylsiloxane (PDMS) are used as transfer medium. **e** Image of multilayer MoS₂ films with precise-controlled twist angles on Si substrates with 300 nm SiO₂. Source data are provided as a Source Data file.

exciton mapping on a $100 \times 100\ \mu m^2$ area (right inset of Fig. 2e) shows that our fabrication process provides high-quality twisted MoS₂ homostructures, which is further supported via the laser scanning confocal fluorescence microscopy image (left inset in Fig. 2e). 0° bilayer samples show the same uniformity (Supplementary Fig. 7). In Supplementary Fig. 8, the PL spectra of top and bottom monolayer are identical, indicating that our transfer method would not damage MoS₂ films. For more information, please see Supplementary Notes 5 and 6.

**Twist angle-dependent spectral properties of twisted multilayer MoS₂ films.** Since a series of MoS₂ films with accurately controlled twist angles are available, we thus performed Raman and PL to characterize these large-area samples. Figure 3a is PL spectra of twisted bilayer MoS₂ films. To highlight the exciton of indirect bandgap, signal intensities between 706 nm to 950 nm are multiplied by 7. Both the intensity and position of A and B excitons peaks barely change with twist angles. The energies of A (B) exciton is around 1.86 eV (2.01 eV), indicating the spin-orbit coupling is 0.15 eV, which is in a good agreement with previous theoretical and experimental results[35,36]. In contrast, the position of indirect bandgap exciton peaks shows a clear blue shift and the intensity of these peaks increases with the twist angle, confirming the previous study[31]. The energy of indirect exciton exponentially increases from 1.44 to 1.63 eV as the twist angle increases from 0° to 30° (Fig. 3b). Such twist angle-dependent energies of indirect exciton stem from that the interlayer coupling decrease with increasing the twist angle, which leads to the energy of critical points **Q** (**Γ**) upshift (downshift)[31,36–38]. We also applied PL characterizations of our twisted trilayer MoS₂ films. Consider the

complexity of the trilayer structure, we have prepared only five samples with representative stacked configuration, as shown in Fig. 1e. PL spectra (Fig. 3c) and excitons' energy (Fig. 3d) of twisted trilayer MoS₂ films also indicate that the energies of indirect exciton can be tuned by precise control of the twist angle of each layer. Besides, compared to natural bilayer and trilayer MoS₂ in Supplementary Fig. 9, our artificial bilayers and trilayer show similar spectral properties. In other words, we can realize specific electronic bands by twisted layers of multilayer MoS₂ films.

Raman spectra of twisted multilayer MoS₂ films are shown in Fig. 4. Figure 4a shows the Raman spectra of twisted bilayer MoS₂ films. Two prominent peaks around 384 and 407 cm⁻¹ can be seen, originating from the in-plane $E_{2g}$ and out-of-plane $A_{1g}$ modes at Brillouin-zone center of the monolayer constituent, respectively[39]. The position of $E_{2g}$ peaks is not sensitive to twist angles, while the $A_{1g}$ peaks shift with the twist angles (Fig. 4b), being consistent with the previous results[31]. This distinct angle-dependence is due to that $E_{2g}$ and $A_{1g}$ modes are mainly determined by long-range Coulombic interlayer interactions and interlayer vdW interactions, respectively[40,41]. The softening of $A_{1g}$ phonon with increasing twist angle indicates that the interlayer coupling is strongest for the 0° twist angle[31]. Apart from these two modes, we can observe a mode at about 411 cm⁻¹ when the twist angle is larger than 8° (Fig. 4a). This Raman mode can be assigned as Moiré phonon related with the $A_{1g}$ phonon branch (here we denote it as F$A_{1g}$), which stems from the off-center phonons of monolayer linked with the lattice vectors of Moiré reciprocal space[42]. F$A_{1g}$ peaks exhibit a sine-like behavior with twist angle (Fig. 4b). Since the Moiré phonons F$A_{1g}$ at

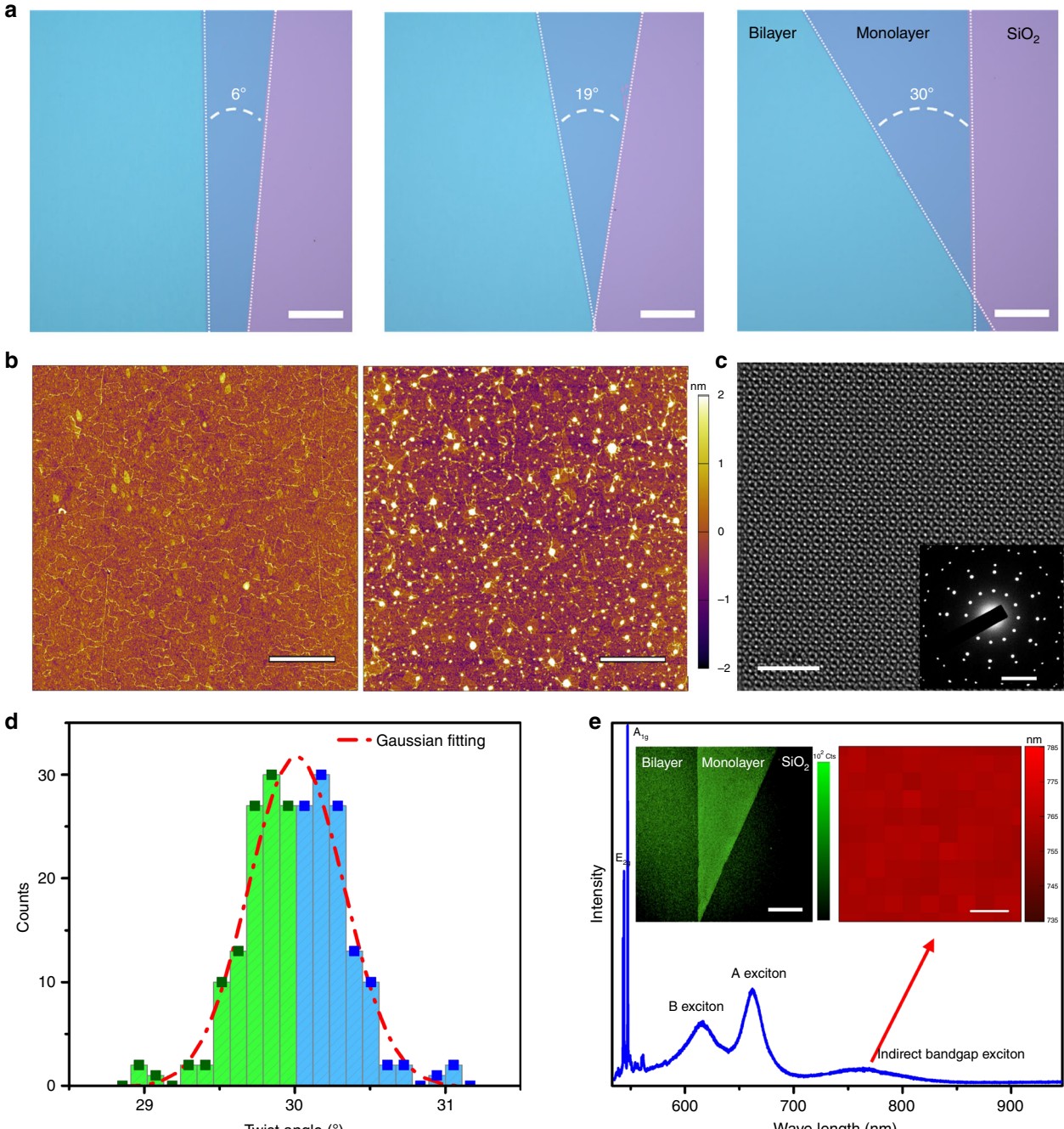

**Fig. 2 High-quality twisted bilayer MoS₂ films. a** Optical Images of three typical transferred twisted bilayer MoS$_2$ films on Si substrates with 300 nm SiO$_2$: 6°, 19°, and 30°, scale bar 300 μm. **b** AFM images of the transferred monolayer (left) and 30° bilayer (right) MoS$_2$ films, scale bar 2 μm. **c** STEM image after FFT filtering of 30° stacked bilayer MoS$_2$ film, scale bar 3 nm; insert is electron diffraction pattern of 30° stacked bilayer MoS$_2$ film, scale bar 5 nm$^{-1}$. **d** Twist angle distribution of eight different 30° stacked bilayer MoS$_2$ film samples, red dash line is the Gaussian fitting. Blue region is just a copy of the green region, to make the chart symmetric. **e** PL spectrum of 30° stacked bilayer MoS$_2$ film. Left inset in **e** is the laser scanning confocal fluorescence microscopy image, scale bar 300 μm; the right inset is a 100 × 100 μm$^2$ mapping of the indirect bandgap position, scale bar 20 μm. Source data are provided as a Source Data file.

different twist angles are derived from distinct different wave vectors of the phonon dispersion, it provides an effective way to map the phonon dispersions[42]. In addition to the phonon energies, the intensities of Raman peaks are also dependent on the twist angle, as shown in Fig. 4c. The intensities of $E_{2g}$ and $A_{1g}$ peaks linearly decrease from 0° to 30° with a larger slope for $A_{1g}$ mode. This can be understood as that $A_{1g}$ mode possesses a stronger electron-phonon coupling than the $E_{2g}$ mode[43]. In

contrast, the intensities of F$A_{1g}$ exponentially increase from 8° to 30°, resulting from the sharply increasing density of Moiré phonon.

To further confirm the good interfacial coupling of our samples, we performed the Raman spectra in the ultralow-frequency region, which provides a fingerprint for benchmarking the quality of interfacial coupling[15,44]. The shear (S) and layer-breathing (LB) modes of twisted bilayer MoS$_2$ homostructures are

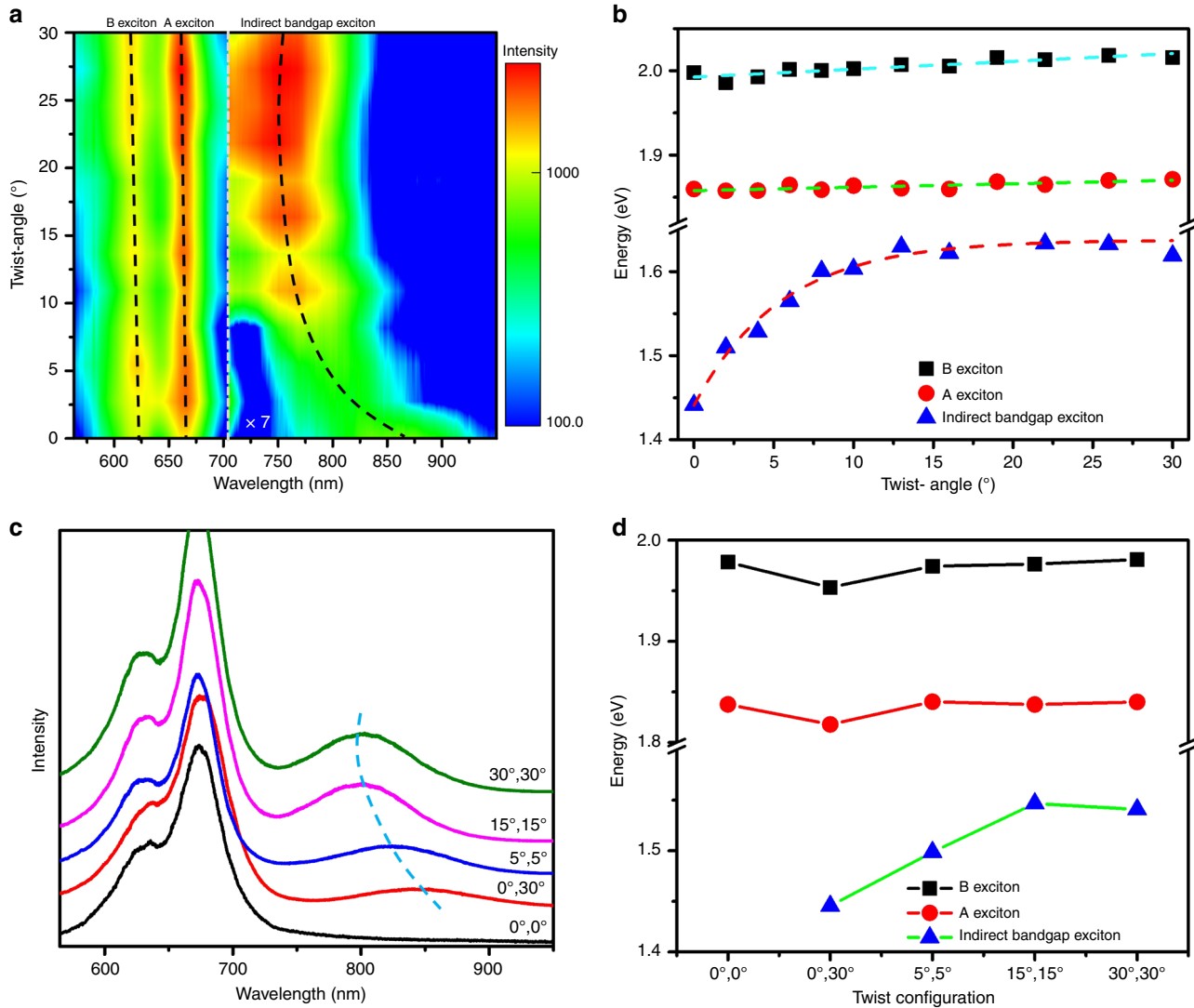

**Fig. 3 PL spectra characterization of twisted multilayer MoS₂ films. a** PL spectra of twisted bilayer MoS₂ films as a function of twist angle; the signal intensity between 706 nm to 950 nm is multiplied by 7. **b** Excitons' energy as a function of the twist angle; dash lines are linear (A and B excitons) and exponential (indirect bandgap exciton) fitting. **c** PL spectra of twisted trilayer MoS₂ films with various twist configurations. **d** Excitons' energy as a function of twist configuration. Source data are provided as a Source Data file.

shown in Fig. 4d, together with the data from monolayer MoS₂. For 0° twisted bilayer sample, both the S and LB modes are observed and located around 22 and 38 cm⁻¹, respectively, in agreement with the signatures of exfoliated[45] or CVD-grown bilayer samples[46]. This demonstrates that the interlayer coupling of our samples is quite strong. For other twist angles, the S modes are missing and LB modes redshift, indicating the weakening of interlayer coupling[15,42]. Figure 4e presents the Raman spectra of five twisted trilayer MoS₂ samples. Being akin to the results of the bilayer, we can observe not only the in-plane $E_{2g}$ and out-of-plane $A_{1g}$ modes, but also the Moiré phonons F$A_{1g}$. Strikingly, the effect of Moiré phonons in trilayer twisted MoS₂ with structure (0°, 30°) and (30°, 0°) is stronger than that of bilayer samples, due to the larger density of moiré phonon in trilayer MoS₂ films.

**Device characterization of twisted multilayer MoS₂ films.** Finally, we investigated the electrical properties of twisted multilayer MoS₂ films. We used a standard ultraviolet (UV) lithography method and deposited Ti/Au as electrodes. Inset of Fig. 5a

shows the optical image of our device array made from 30° bilayer sample. Devices exhibit high quality and integrity. Standard $I/V$g curves are shown in Fig. 5a. Figure 5b is $I/V$ curves of our device, which are linear under different back gate voltages, showing good contact between MoS₂ and metal electrodes. Figure 5c is the on/off ratio statistics of 0° and 30° stacked bilayer MoS₂ devices, on/off ratio of our 30°/0° stacked bilayer MoS₂ devices device is ~10⁸/10⁷. 30° stacked bilayer MoS₂ devices have higher on/off than 0° is due to 30° stacked bilayer MoS₂ devices have higher on-current (Supplementary Fig. 10). Figure 5d is the electron mobility of device arrays made from MoS₂ films with different stacking sequences and interlayer twist angles. We can see that 30° twisted structure have higher mobility than 0° twisted structure, as electron mobilities of 30°/(30°,0°) twisted samples are higher than that of 0°/(0°, 0°) and (0°, 30°). We attribute the electron mobility of 30° bilayer higher than that of (30°,0°) and (0°, 30°) trilayer to the screen of the electric field by the bottom MoS₂ layers and the enhanced scattering effect of by trapped interlayer bubbles in trilayer samples. The higher mobility of 30° twist angle may be due to the interlayer decoupling by incommensurate structure or

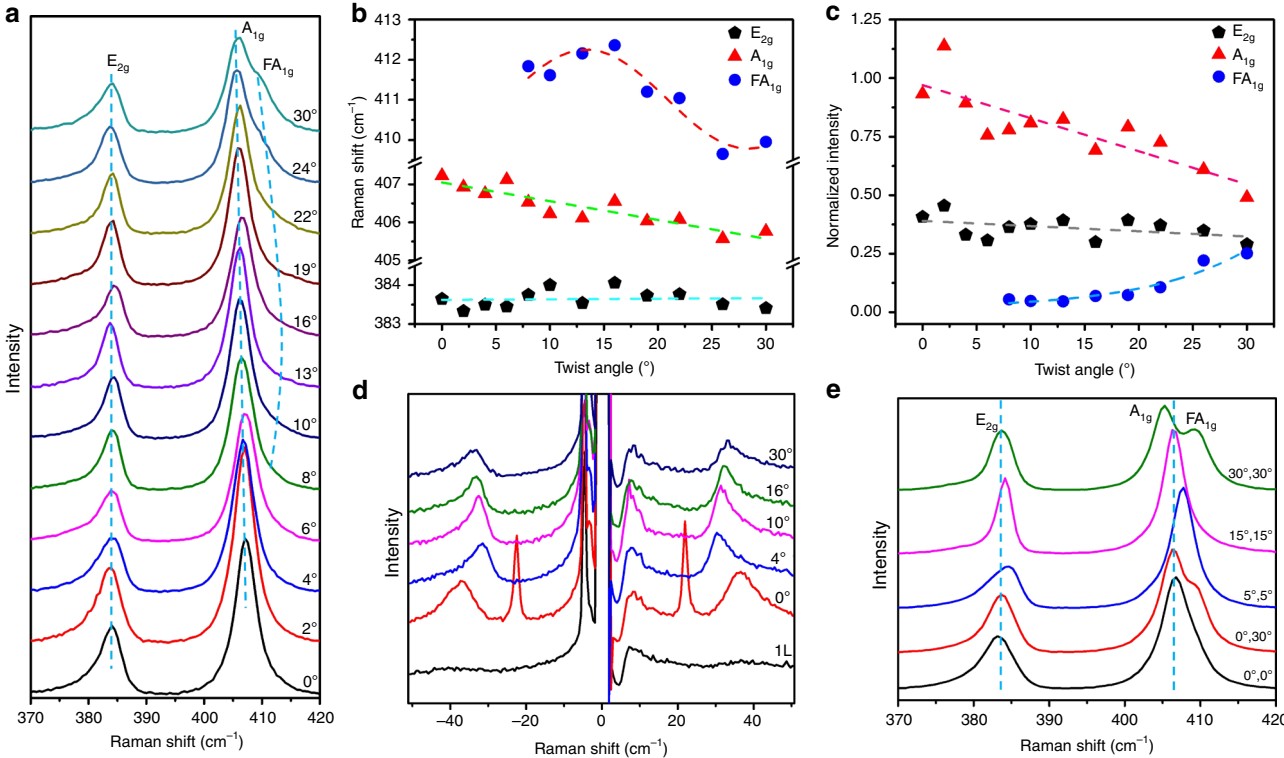

**Fig. 4 Raman characterizations of the twisted bilayer and trilayer MoS₂ films. a** Raman spectra of a series of transferred bilayer MoS₂ films with controlled twist angle, each Raman spectrum was calibrated and normalized by the position and intensity of silicon peak at 520.7 cm⁻¹. **b** The position of $E_{2g}$, $A_{1g}$, and $FA_{1g}$ Raman peaks as a function of twist angle, dash lines are linear ($E_{2g}$, $A_{1g}$) and sinusoidal ($FA_{1g}$) fitting. **c** The intensity of $E_{2g}$, $A_{1g}$, and $FA_{1g}$ Raman peaks as a function of twist angle, dash lines are linear ($E_{2g}$, $A_{1g}$) and exponential ($FA_{1g}$) fitting. **d** Low-wavenumber Raman spectra of monolayer and bilayer twisted MoS₂ films. **e** Raman spectra of trilayer twisted MoS₂ films with different twist configuration. Source data are provided as a Source Data file.

smaller interlayer resistance[47]. For more information, please refer to Supplementary Notes 7.

## Discussion

In conclusion, we successfully obtained large-scale MoS₂ homo-structures with precise-controlled layer number and interlayer twist angle by using an advanced epitaxial growth method and water assisted transfer method. Our results show that the inter-layer twist angle of MoS₂ films has a strong influence on both the spectroscopic properties and electronic mobility. Our work pro-mises a cost-effective and scalable process to prepare large-scale vdW homo- and hetero-structures with precise controlling the twist angle, which would open a pathway to industrial applica-tions of twistable electronics and photonics.

## Methods

**Epitaxial CVD growth of Wafer-Scale MoS₂.** The MoS₂ growth[28] was performed in our home-made three-temperature-zone chemical vapor deposition (CVD) system. S (Alfa Aesar, 99.9%, 4 g) and MoO₃ (Alfa Aesar, 99.999%, 50 mg) pow-ders, loaded in two separate inner tubes, were used as sources and placed at Zone-I and zone-II, respectively, and 2 inches. Sapphire wafers were loaded in zone-III as the substrates. During the growth, Ar (gas flow rate 100 sccm) and Ar/O₂ (gas flow rate 75/3 sccm) were used as carrying gases. The temperatures for the S source, MoO₃ source, and wafer substrate are 115, 530, and 930 °C, respectively. The growth duration is ~40-min, and the pressure in the growth chamber is ~1 Torr.

**Sample characterizations.** AFM imaging was performed by Veeco Multimode III and Asylum Research Cypher S. Raman and PL characterizations were carried out on a Horiba Jobin Yvon Lab RAM HR-Evolution Raman confocal microscope with an excitation laser wavelength of 532 nm, a laser power of 100 μW. LEED mea-surement was performed in UHV chambers at a base pressure of <1.0 × 10⁻¹⁰

mbar. Samples were annealed in the UHV chamber at 200 °C for 2 h. For LEED, the electron beam energy ranges from 100 to 200 eV. SAED was performed in a TEM (Philips CM200) operating at 200 kV. Atomic-resolution HAADF-STEM images were acquired by an aberration-corrected Nion U-HERMES200 system operated at 60 kV.

**Transfer methods.** PDMS films used in the transfer process were prepared using SYLGARD 184 (Dow Corning Corporation), a two-part kit consisting of pre-polymer (base) and cross-linker (curing agent). We mixed the prepolymer and cross-linker at a 10:1 weight ratio and cured the cast PDMS films on silicon wafers at 100 °C for 4 h. During the transfer process, the PDMS/MoS₂ films were clamped by a manipulator equipped on home-made step-motor linear guides for assisting both their peeling-off from sapphire substrates and stamping onto receiving sub-strates, same with our previous work[28]. After transfer, all samples were annealed at 400 °C for 8 h, under the protection of 20 sccm H₂/150 sccm Ar gas, ~1 Torr. Diagram of the transfer process, please see Supplementary Fig. 3.

**FET device fabrication and measurements.** The transferred twisted multilayer MoS₂ films were firstly patterned with RIE (Plasma Lab 80 Plus, Oxford Instruments Company) by oxygen plasma, and then the standard UV-lithography (MA6, Karl Suss) process was used to pattern source/drain contacts with AR-5350 as the photoresist, which was spin coated on sample surface at 4000 rpm and baked at 100 °C for 4 min. The developer is AR 300-47. 2/30 nm Ti/Au contacts were deposited by home-made e-beam evaporation system. The elec-trical measurements were carried out in a Janis vacuum four-probe station with Agilent semiconductor parameter analyzers (1456C and B1500) under a base pressure of 3 × 10⁻⁶ mbar.

## Data availability

The authors declare that the data supporting the findings of this study are available within the paper and its supplementary information files. The source data underlying Figs. 1b, c, 2b–d, 3a–e, 4a–d and 5a–d, and Supplementary Figs. 2b–e, 4a–d, 6f, 7b–d, 8, 9b, c and 10 are provided as a Source Data file.

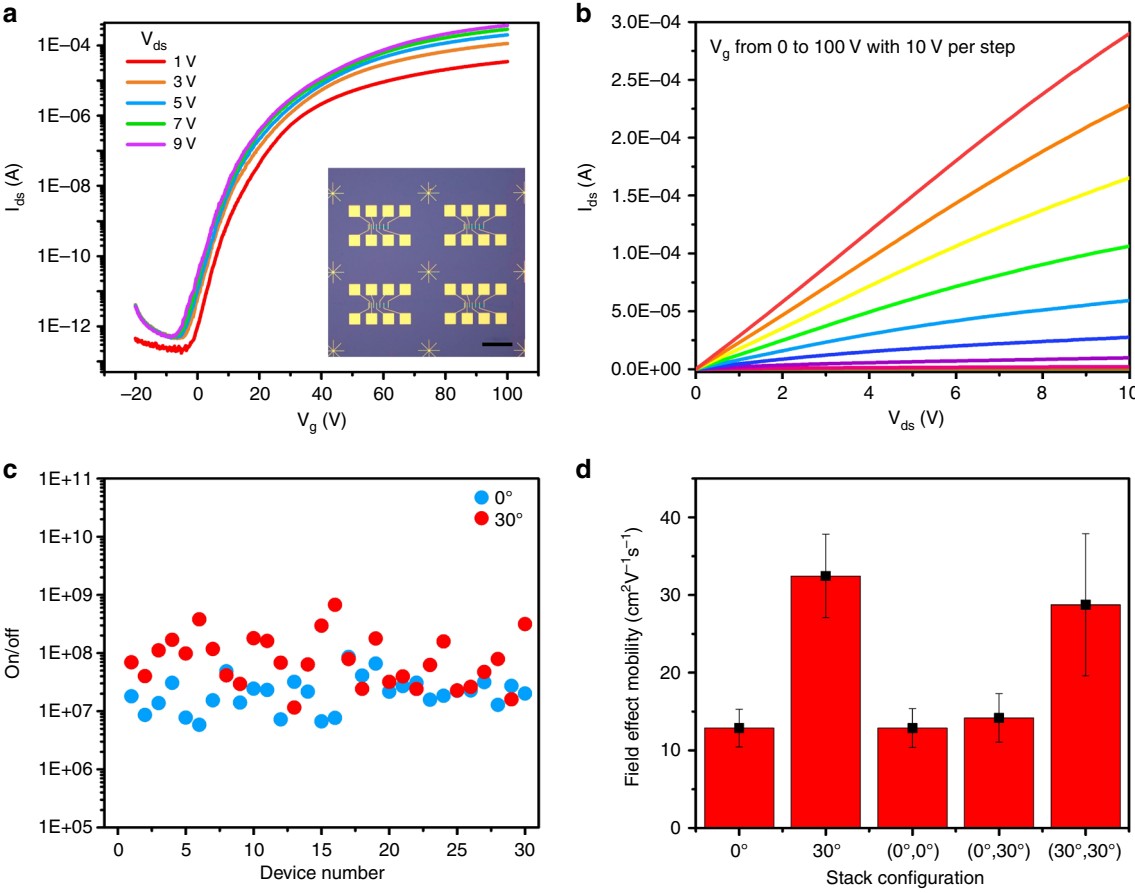

**Fig. 5 Electrical properties of twisted multilayer MoS₂ films. a** Electrical transfer curves of a typical 30° twisted bilayer MoS₂ FET, inset is an optical image of a device array, scale bar 400 μm. **b** Electrical output curves of a typical 30° twisted bilayer MoS₂ FET. **c** On/off ratio of 0° and 30° devices. **d** Mobility statistics of twisted multilayer MoS₂ films, error bars are Standard Deviations. Source data are provided as a Source Data file.

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

## Acknowledgements

This project was supported by the National Science Foundation of China (NSFC, Grant Nos. 61734001, 11834017, 11574361, and 51572289), the Strategic Priority Research Program (B) of CAS (Grant No. XDB30000000), the Key Research Program of Frontier Sciences of CAS (Grant No. QYZDB-SSW-SLH004), the National Key R&D program of China (Grant No. 2016YFA0300904), and the Youth Innovation Promotion Association CAS (No. 2018013). T.P. acknowledges support from the project CZ.02.1.01/0.0/0.0/15_003/0000464. S.Z. acknowledges support from the project Academy of Finland (Grant Nos. 295777, 312297, and 314810), Academy of Finland Flagship Program (Grant No. 320167, PREIN), the European Union's Horizon 2020 research and innovation program (Grant No. 820423, S2QUIP), and ERC (Grant No. 834742)

## Author contributions

G.Z. and Y.R. supervised the research; M.L. design the experiment, preformed sample preparing & characterization; Z.W. performed the growth of MoS$_2$ and electrical characterization. L.D. helped analysis spectra data. Q.Q. and Y.H. performed the growth of MoS$_2$; F.W. and J.Z. helped sample preparation. X.X. and K.L. performed LEED; J.T., B.H., and P.G. performed TEM; T.P., Z.S., and G.Z. revised manuscript; M.L., L.D., and Y.R. wrote the article. All authors commented on the manuscript.

## Competing interests

The authors declare no competing interests.
