## [Peer Review File · Nature Communications]

Reviewers' comments:

Reviewer #1 (Remarks to the Author):

The manuscript reports the precise control of twist angle in stacked multilayer MoS₂ via water-assisted transfer of highly-oriented monolayer CVD grown MoS₂. They can realize the large-scale twisted MoS₂ fabrication with specific twisted angle by large-scale growth and the following transfer. They find that the twist angle can continuously change the indirect band gap of stacked multilayer MoS₂ homostructures and the electrical properties. Overall, I think this is a great work and an important advance for the fabrication of twistronic devices. Most importantly it is an approach that is likely to be scalable. I have a few concerns which I am listing below in detail. I think that the authors should address these concerns before this manuscript is accepted at Nature Communications.

1. The author claimed the CVD grown MoS₂ on sapphire is highly oriented. I think the LEED results are not strong enough to demonstrate the orientation with 0 and 60 degrees. What's the intermediate growth morphology of MoS₂? How do the MoS₂ flakes merge together?
2. The quality of MoS₂ is easy to be degraded during transfer. Can the author perform the PL spectra of bottom and top layer to demonstrate the same nature of MoS₂ after transfer?
3. The author reports that the stack structure can affect the electrical properties. Does the on/off ratio have some change with the stack angles?
4. Can the author present the PL spectra of monolayer and bilayer MoS₂ in the same figure, which could help to compare the change before and after stack?
5. Even though the as-grown CVD MoS₂ is highly oriented, there should have a certain amount of grain boundaries when merging together. The grain boundaries should have some influence on the PL. Both as-grown MoS₂ and as-transferred MoS₂ results in this paper seem perfect. I wonder if there are some wrinkles and defects, which would influence the PL spectra.
6. The as-grown CVD MoS₂ film is consisted of MoS₂ with 0 degree and 60 degree. Thus, the bilayer MoS₂ should exist in the mixed AA and AB structures. But in this paper, the AA and AB structures and their optical properties are not in consideration. The author should address the concern about the AA and AB stacked MoS₂.
7. The paper presents many optical images of as-transferred MoS₂, which look quite clean. But usually, the contaminations during transfer cannot be seen just by the optical images. Can the author add AFM images to show the clean surface after transfer?

Reviewer #2 (Remarks to the Author):

Liao et al., present a new method to control the angular alignment between layers of a homostructure in the millimeters scale. They use a combination of PDMS and DI water to transfer epitaxially grown MoS₂ layers, in some sort this is the same technique developed by Kim et al., Nano Letters 16 1989 (2016) but this time using large area layers. They use different techniques (Raman spectroscopy, photoluminescence, STEM, LEED) to show the strengths of their technique. I have some questions for the authors:

- What is the precision of the alignment? Can the authors give an estimation of the angles with error bars? What about reproducibility? Is it possible to align two layers to a small angle?
- Concerning their transfer technique: I'm curious to know why the MoS₂ layer adhesion to PDMS is stronger than to the original substrate (sapphire) and why the adhesion to SiO₂ is stronger than to the PDMS? Or there is an extra parameter (e.g., temperature) that can be used to adjust this adhesion?
- I understand they use the indirect gap signature to show the homogeneity of the transfer, however given that the minimum of elastic energy of a bilayer system will be around zero degrees the homogeneity should be shown for small angles and not for large angles.
- Concerning the PL measurements where the indirect bandgap exciton is observed I believe a

explanation is necessary. For example in reference 31 of the manuscript I found a simple but clear explanation of the decrease of the intensity of this peak with layer alignment. I consider this explanation necessary for the non-experts such as myself. I found the explanation "Such twist angle-dependent energies of indirect exciton stem from that the interlayer coupling decrease with increasing the twist angle which leads to the energy of critical points Q (Γ) upshift (downshift)" a bit rough and only meant for specialist.

- It will be good to shown in the case of bilayer and trilayer the PL response for "naturally stacked" high quality systems in order to compare with the state of the art.

- In the Raman spectrum if the FA_1g mode is associated with the moiré phonons mode Why does it appears only at 8 degrees when the moiré is already small and not closer to zero degrees? Is this reason why in the trilayer this FA_1g mode gets stronger? Because two of the layers are oriented close to 0 degrees

- The statement : "We can see that 30° twisted structure have higher mobility than 0° twisted structure," is not really accurate when the difference is very small and it could even be between the error bars.

- The authors do not discuss possible water contamination that could give rise to the well known problem of bubbles trapped between the layers.

I consider that the authors should address the questions above and that they should make clear the reproducibility and accuracy of their method, which seems to me is their strongest point. What concerns to their scientific results they seem correct and their interpretation sound, however I have a hard time making a difference with the results already shown in reference 31. Therefore, I left the decision about publication to the editor, which I guess will be based on the interest of the readers.

Additional comments:

The statement: "Fig. 1a and Fig. S1 are optical images of the as-grown monolayer MoS2 film on a 2-inch sapphire wafer, showing that the film is highly uniform and covers the entire wafer." Seems incorrect to me given that it is not possible to see the uniformity of the film. I recommend to erase this confusing sentence.

Same with the sentence "The optical images indicate that the surfaces of all transferred films are clean and uniform." This is misleading given that most of the contamination for 2D materials is not visible in optical microscopes.

Scales of figure 2d are difficult to read

Reviewer #3 (Remarks to the Author):

The manuscript describes the sequential transfer of grown MoS2 monolayers into angle controlled bi- or tri-layer stacks. The authors employ a previously reported growth technique, which produces aligned but not single crystal MoS2 monolayer (there are in fact a lot of domain boundary defects, as mentioned in their prior paper but not here). There have been previous reports on aligned stacking of exfoliated flakes, here the authors do this with the grown mono-layer.

A major claim is the "precisely" or "accurately" controlled angle, which I do not at all find substantiated. Too little detail is given about the transfer methodology and set-up. It is not clear what max. area of transfer can be achieved. Looking at Fig. 2a, unlike claimed, the edges of each layer are not sharp, and the authors must present a statistical discussion and error in alignment. No error bars at all are given, neither is total error of alignment. This is not the standard one would expect to see in Nature journal. I note the error bar discussion needs also to include each characterisation technique, and characterisation here needs to cover the claimed larger areas to mm size.

Another claim is that interface of transferred films is "ultraclean". Again this is not substantiated at all. The very small area moire pattern from TEM analysis is certainly not proving anything in this regard. All analysis (incl. the Raman spectra, no maps? Optical maps clearly not sufficient as even dirty films can look homogeneous as is well known) is done over very small (in comparison to wafer growth and mm cuts) areas, even though claim is of scaled, larger sample capability. No (statistically relevant) evidence is provided. I expect to see analysis over whole at least mm-scale. No comments on possible PDMS residual or any other contamination, that can be expected for "wet" processing. Why should the technique give "ultra-clean" interface over mm areas?

The data in Fig. 5 b is illegible. Not sufficient detail on mobility analysis is given (at least in SI). Not clear what the error bars refer to in Fig 5c. What is expected contact resistance?

I expect also discussion about effect of the many domain boundaries in the films used, as for instance such boundaries are known to trap contamination.

Overall, it is current state I can not recommend this manuscript for publication in Nat. Comm. There is insufficient data to support any of the major claims. I would recommend the authors to refocus on all the details of the aligned transfer technique (incl. careful error analysis), as there is currently a lot of interest in the community on that. But solid data is required not exaggerated claims.

Point-by-point reply to referees' comments and list of changes

Reviewer #1

The manuscript reports the precise control of twist angle in stacked multilayer MoS₂ via water-assisted transfer of highly-oriented monolayer CVD grown MoS₂. They can realize the large-scale twisted MoS₂ fabrication with specific twisted angle by large-scale growth and the following transfer. They find that the twist angle can continuously change the indirect band gap of stacked multilayer MoS₂ homostructures and the electrical properties. Over all, I think this is a great work and an important advance for the fabrication of twistrionics devices. Most importantly it is an approach that is likely to be scalable. I have a few concerns which I am listing below in detail. I think that the authors should address these concerns before this manuscript is accepted at Nature Communications.

We thank the reviewer for the positive comments “I think this is a great work and an important advance for the fabrication of twistrionics devices. Most importantly, it is an approach that is likely to be scalable”. The constructive comments made by the reviewer are addressed below.

Comments #1: The author claimed the CVD grown MoS₂ on sapphire is highly oriented. I think the LEED results are not strong enough to demonstrate the orientation with 0 and 60 degrees. What's the intermediate growth morphology of MoS₂? How do the MoS₂ flakes merge together?

Author response: In our previous work ref. 26 [ACS Nano 11, 12001 (2017)], we did detailed characterizations for the crystal orientation of as-grown MoS₂ films. In Fig. 2 of ref. 26, we used LEED, HRTEM, ARPES and atomic resolution AFM (stick-slip lattice images) to demonstrate the highly oriented nature of our films. In LEED characterizations, we only saw one set of diffraction spots on 2cm*2cm scale film (please see the recorded video in the supporting info. of ref. 26); In HRTEM we only found twin boundaries; The AFM lattice images of 100 different domains show that the orientation error is within 3.5°, mainly due to the thermal drift of the machine. Also, no triple boundary junction was observed, which means only two orientations. The key point to have highly oriented MoS₂ films is to grow under relatively high temperatures. In Fig. S10 of ref. 26, we showed that when raising the growth temperature from 770° to 930°, the second set of LEED spots, which means another orientation, disappeared. The diffraction spots also become sharper under high growth temperature, indicating the enhancement of orientation. All the characterizations showed strong evidence that only 0° and 60° MoS₂ domains exist..

In Fig. S4 of ref.26, we discussed the intermediate growth morphology. First, MoS₂ islands were formed on the sapphire surface, and then expanded and merged. If two domains have the same orientation, they will merge without interior boundary; otherwise, there will be twin boundaries.

Below are the Figures of ref.26.

Figure 2. Lattice alignment between as-grown MoS₂ and sapphire substrates. (a) LEED pattern of monolayer MoS₂ on sapphire and the lattice orientation of sapphire wafer; the incident electron energy is 147 eV. The zigzag and armchair directions of MoS₂ were parallel to $[\bar{1}\bar{1}20]$ and $[\bar{1}100]$ directions of sapphire, respectively. The orientation of MoS₂ lattice aligned with sapphire. (b) ARPES spectra of the as-grown monolayer MoS₂. (c) The HRTEM image of the stitched domain boundary in monolayer MoS₂. The bright spots correspond to Mo atoms, and the dim spots correspond to S atoms. The dashed line shows the domain boundary, the two yellow triangles show that the angle between the two domains is 60°. (d) Selected area electron diffraction (SAED) pattern; the diaphragm is $\sim 1 \mu\text{m}$. (e) Orientation distribution of 100 different sample points homogeneously distributed in $20 \mu\text{m} \times 20 \mu\text{m}$ square. (f) Schematic illustration of the film stitched by I- and II-domains

Fig. S4. (a) AFM image of MoS₂ islands on sapphire wafer (after 30min growth), the grain size is about $1 \mu\text{m}^2$. (b) AFM image of continuous MoS₂ film on sapphire wafer (after 40 min growth). (c) AFM image of continuous MoS₂ film on sapphire wafer (after 60 min growth). (d) Coverage ratio for monolayer MoS₂ as a function of growth time.

Fig. S10. LEED images of MoS₂ films on sapphire with different growth temperatures.

Comments #2: The quality of MoS₂ is easy to be degraded during transfer. Can the author perform the PL spectra of bottom and top layer to demonstrate the same nature of MoS₂ after transfer?

Author response: We measured the PL spectra of a 0° stacked bilayer sample, the data is as below. From which we can see, the PL spectra of top and bottom layers are almost identical. So our transfer method would not damage the MoS₂ films. We have added this data in supplementary materials, as Fig. S8. And in main text line 134 page 5, we add “In Fig. S8, the PL spectra of top and bottom monolayer are identical, indicating that our transfer method would not damage the MoS₂ films.”

Comments #3: The author report that the stack structure can affect the electrical properties. Does the on/off ratio have some change with the stack angles?

Author response: Thank you for your comment. We have fabricated some new devices to make sure the electrical properties of the stack structures. The electrical data is as below. The result is consistent with our previous work. From Fig. c and d, we can see the 30° bilayer sample has higher on/off ratio and mobilities. Fig. 5 has been replaced, and we add “ Fig. 5c is the on/off ratio statistics of 0° and 30° stacked bilayer MoS₂ devices, on/off ratio of our 30°/0° stacked bilayer MoS₂ devices device is ~ 10⁸/10⁷. 30° stacked bilayer MoS₂ devices have higher on/off than 0° is due to 30° stacked bilayer MoS₂ devices have higher on-current (Fig. S10).” to line 231 page 10.

Comments #4: Can the author present the PL spectra of monolayer and bilayer MoS₂ in the same figure, which could help to compare the change before and after stack?

Author response: Thanks for the suggestion. Please refer to the image of Comments #2, which shows a very clear intensity and shape change of the PL spectra after stack.

Comments #5: Even though the as-grown CVD MoS₂ is highly oriented, there should have a certain amount of grain boundaries when merging together. The grain boundaries should have some influence on the PL. Both as-grown MoS₂ and as-transferred MoS₂ results in this paper seem perfect. I wonder if there are some wrinkles and defects, which would influence the PL spectra.

Author response: We thank the reviewer for the professional comments. Yes, the as-grown CVD MoS₂ has a certain amount of twin grain boundaries. And the twin grain boundaries can affect the PL spectra and usually causes a decrease in PL intensity. Since the size of our laser spot (~ 5 μm²) is larger than the grain size (~ 1 μm²) and twin grain boundaries are uniformly distributed in the sample, the effects of grain boundaries cannot be observed. Therefore, we can obtain a uniform result.

Comments #6: The as-grown CVD MoS₂ film is consisted of MoS₂ with 0 degree and 60 degree. Thus, the bilayer MoS₂ should exist in the mixed AA and AB structures. But in this paper, the AA and AB structures and their optical properties are not in consideration. The author should address the concern about the AA and AB stacked MoS₂.

Author response: Yes, you are right. The bilayer MoS₂ should exist in the mixed AA and AB structures as the as-grown CVD MoS₂ film consists of MoS₂ with 0 and 60 degrees. However, the optical properties between AA and AB stacking are the same because both the interlayer coupling

and interlayer distance of these two structures are almost the same (Nat. Commun. 5, 4966 (2014), Ref. 31). Therefore, we mark both the AA and AB stacked MoS₂ as 0° MoS₂ bilayer in our manuscript.

Comments #7: The paper presents many optical images of as-transferred MoS₂, which look quite clean. But usually, the contaminations during transfer cannot be seen just by the optical images. Can the author add AFM images to show the clean surface after transfer?

Author response: Thanks for your suggestions. We agree that optical images are not enough to demonstrate the “cleanness” of our samples. So, to demonstrate the clean surface and interface, we scanned all the surfaces involved in the transfer process. Images are shown as below. Fig. **a-f** are AFM images of **(a)** PDMS surface; **(b)** as grown MoS₂/sapphire surface; **(c)** MoS₂/sapphire surface after stamping and peeling off by PDMS in air; **(d)** MoS₂/PDMS surface; **(e)** surface of transferred monolayer MoS₂ on Si substrate with 300nm SiO₂; **(f)** surface of transferred 30° bilayer MoS₂ on Si substrate with 300nm SiO₂. From Fig. **a**, we can see the surface of PDMS is flat and clean, Ra=677.3pm, no contamination observed. Fig. **b** and **c** show that after PDMS stamping and peeling off in the air, nothing remains on MoS₂/sapphire sample surface, even previous contaminations on twin boundaries disappear. Thus, PDMS will not induce any contamination to MoS₂ surfaces (even can remove contaminations), it is “inert”. Fig. **d** shows that after peeling off MoS₂ from sapphire by PDMS in DI water, the MoS₂/PDMS surface is clean, which means no contamination will be induced during the water assistant transfer process. As illustrated in Fig. **e**, after annealing, the surface of transferred monolayer MoS₂ film is clean and flat. For the bilayer sample in Fig. **f**, we can see most of the area is flat, only a few small bubbles exist (maximum 10%). Bilayer sample has more bubbles than monolayer is due to that trapped gas cannot escape as interlayer coupling between two MoS₂ layer is much stronger than that between MoS₂ and polycrystal SiO₂. These AFM images demonstrate that our samples have bot clean interfaces and surfaces. We put this Figure as Fig. S4 and add “*AFM images in Fig. S4a-d indicate that all the surfaces involved in the transfer process are clean and flat, no contamination was observed. Fig.S4e and f show that the surface of transferred monolayer MoS₂ film is also clean and flat, and bilayer sample only has few bubbles (maximum 10% of the area), indicating the high quality of our samples.*” to 115 line page 5.

Reviewer #2

Liao et al., present a new method to control the angular alignment between layers of a homostructures in the millimeters scale. They use a combination of PDMS and DI water to transfer epitaxial grown MoS₂ layers, in some sort this is the same technique developed by Kim et al., Nano Letters 16 1989 (2016) but this time using large area layers. They use different techniques (Raman spectroscopy, photoluminescence, STEM, LEED) to show the strengths of their technique.

We appreciate these comments.

Comments #1: What is the precision of the alignment? Can the authors give an estimation of the angles with error bars? What about reproducibility? Is it possible to align two layers to a small angle?

Author response: We want to demonstrate the precision and the reproducibility of our method from two parts.

First is the method. The linear guide we used to cut the sapphire has maximum parallel misalignment about 20μm/100mm, which will induce a maximum 0.01° error. We also changed the diamond head to a tungsten needle and drew a line on MoS₂/sapphire sample surface to further test the precision of the linear guide. Coordinates of points on the line were measured by Nikon measuring microscope MM-200. The consequence is as Fig. **a-b** below. For a 30mm range, the misalignment is below 0.005mm, which means a 0.0095° error, agree well with the precision of the linear guide. The rotational stage we used has the resolution 0.033° (1/30°), which can provide a relatively precise twist angle between two stack layers.

Second is to directly measure the twist angle of our bilayer samples by TEM electron diffraction from different regions. We tested three 30° twisted bilayer samples. As shown below, Fig. a is the optical image of bilayer MoS₂ on a microgrid; Fig. b-e are TEM images and electron diffraction patterns from different regions of three 30° twisted bilayer samples; Fig. f is the statistics of twist angle. We transferred all twist angles within 30° (red region), and the blue region is just a copy of the red region to make the chart symmetric. From the statistics, we can see the error is around 0.4°. From some electron diffraction, we can see patterns are not perfectly symmetric. This may be due to the strain-induced during the TEM sample preparation, misalignment between the electron beam and c axis of the sample, and wrinkles, which will cause extra errors. So we can safely claim the error of our method is below 0.4°. We also revised the Fig. 2 and put this statistic into it. And we add “Fig. 2c is the twist angle distribution of three different 30° stacked bilayer MoS₂ films, measured by TEM electron diffraction from different regions. From the statistics, we can see the error is around 0.4°. Because misalignment between the electron beam and the c axis of sample, strain, and wrinkles will increase error, we can safely claim that the error of our method is below 0.4°. Please refer to supplementary note 4 for more details.” in line 124 page 5. Please refer to the main text.

It is possible to align two layers to a small angle. In the main text, we show a 2° bilayer sample. But consider the error above, the actual angle of our sample should be $2^\circ \pm 0.4^\circ$. We notice the recent works about the “magic angle” of bilayer graphene, which has quite a strict requests of twist angle. We think that there are too many issues that can induce errors during the transfer process, and it is almost impossible to ensure an accurate twist angle of every sample. For our method, if we repeat a lot, we can always find samples to fit some strict requests of small twist angles.

Comments #2: Concerning their transfer technique: I’m curious to know why the MoS_2 layer adhesion to PDMS is stronger than to the original substrate (sapphire) and why the adhesion to SiO_2 is stronger than to the PDMS? Or there is an extra parameter (e.g., temperature) that can be used to adjust this adhesion?

Author response: Actually, both the sapphire and SiO_2 surfaces have stronger adhesion force than PDMS in ambient conditions. The key point to achieve peeling off the MoS_2 films from the sapphire surface is to do it in water. There is a previous report about this phenomenon (*Nano Lett* 10, 1912-1916, 2010.), in which 2D samples can be lifted off from their origin substrates by polymer mediums in water if the original substrate is hydrophilic and the 2D material & polymer mediums are hydrophobic, due to different wettability. In our experiment, the sapphire surface is strong hydrophilic; MoS_2 and PDMS surfaces are relatively hydrophobic. Please refer to the image below:

Comments #3: I understand they use the indirect gap signature to show the homogeneity of the transfer, however given that the minimum of elastic energy of a bilayer system will be around zero degrees the homogeneity should be shown for small angles and not for large angles.

Author response: Thanks for your suggestion. We did $200 \times 200 \mu\text{m}^2$ mappings again on new 0° and 30° samples, the data are as below. (a)/(b) and (c)/(d) are single spectra/mapping of 0° and 30° twisted bilayers. We can see the position of indirect bandgap peaks is still homogeneous. The indirect bandgap peaks' position of our new sample is a little blue shift compare to which in the main text, but the tendency is the same. This may occur due to the different doping levels of different processes of growth (Sapphire wafers offered by two companies). We also put this image into supporting information as Fig. S7 and add " 0° bilayer samples show the same uniformity (Fig.S7)." to line 134 page 5 main text.

Comments #4: Concerning the PL measurements where the indirect bandgap exciton is observed I believe an explanation is necessary. For example, in reference 31 of the manuscript I found a simple but clear explanation of the decrease of the intensity of this peak with layer alignment. I consider this explanation necessary for the non-experts such as myself. I found the explanation" Such twist angle-dependent energies of indirect exciton stem from that the interlayer coupling decrease with increasing the twist angle which leads to the energy of critical points Q (Γ) upshift (downshift)" a bit rough and only meant for specialist.

Author response: We thank the reviewer for pointing this out. The twist angle-dependent energies of indirect exciton are due to that it stems from the Q (Γ) valleys in the conduction (valence) band and the interlayer coupling decreases with increasing the twist angle. Since critical points Q (Γ) possess considerable S-pz characters which lead to significant interlayer hopping, the

energies would upshift (downshift) with increasing the twist angle as the interlayer hopping weakens, being akin to the crossover from the indirect gap in bulk to direct gap in monolayer.

Comments #5: It will be good to shown in the case of bilayer and trilayer the PL response for “naturally stacked” high quality systems in order to compare with the state of the art.

Author response: Thanks for your suggestion. The comparison of exfoliated MoS₂ and stacked MoS₂ is as below. From which, we can see that the “artificial” and “natural” bilayer MoS₂ films have similar peak positions of B, A excitons. The only difference is the intensity. We put this image as Fig. S9 and add “Besides, compared to “natural” bilayer and trilayer MoS₂ in Fig. S9, our “artificial” bilayers, and trilayer show similar spectra properties.” to line 165 page 6 main text.

Comments #6: In the Raman spectrum if the FA_1g mode is associated with the moiré phonons mode Why does it appears only at 8 degrees when the moiré is already small and not closer to zero degrees? Is this reason why in the trilayer this FA_1g mode gets stronger? Because two of the layers are oriented close to 0 degrees

Author response: Since the basic vector of moiré reciprocal lattice is smaller than that of the monolayer, and there are many lattice vectors of moiré reciprocal space in the first Brillouin zone of the monolayer constituent. Therefore, the phonons in the Brillouin zone interior of the monolayer constituent linked with the lattice vectors of the moiré reciprocal space can be folded onto the zone center Γ , and these phonons may become Raman active in the twisted bilayer MoS₂ (called moiré phonons). For different twist angles, the moiré phonons would stem from different positions of the Brillouin zone interior of the monolayer constituent. For a large twist angle, the moiré phonons are folded from the off-center phonons with large lattice vectors in monolayer, which possess large density and strong intensity. For trilayer, the stronger intensity than that of bilayer samples is due to the enhanced density of moiré phonon.

Comments #7: The statement: “We can see that 30° twisted structure have higher mobility than 0° twisted structure,” is not really accurate when the difference is very small and it could even be between the error bars.

Author response: Thanks for your suggestions. We repeated the device fabrication and characterization. The result is as below. As shown in inset of Fig. a, we used the transmission line

structure, so the effect of contact resistance can be excluded. Fig. a and b are electrical transfer and output curves of a standard 30° bilayer device. Fig. c shows the on/off ratio statistics of 0°/30° bilayer devices. Which shows 30° bilayer device have higher on/off ratio. Fig. d is the mobilities of twisted multilayer MoS₂ films, which shows similar phenomenon with pervious devices. We attribute the electron mobility of 30° bilayer higher than that of (30°,0°) and (0°, 30°) trilayer to the screen of electric field by the bottom MoS₂ layers, and the enhanced scattering effect by trapped interlayer bubbles in trilayer samples. Fig. 5 has been replaced, and we revised to line 226-240 page 10 to “Inset of Fig. 5a shows the optical image of our device array made from 30° bilayer sample. Devices exhibit high quality and integrity. Standard I/V_g curves are shown in Fig. 5a. Fig. 5b is I/V curves of our device, which are linear under different back gate voltages, showing good contact between MoS₂ and metal electrodes. Fig. 5c is the on/off ratio statistics of 0° and 30° stacked bilayer MoS₂ devices, on/off ratio of our 30°/0° stacked bilayer MoS₂ devices device is $\sim 10^8/10^7$. 30° stacked bilayer MoS₂ devices have higher on/off than 0° is due to 30° stacked bilayer MoS₂ devices have higher on-current(Fig. S10). Fig. 5d is the electron mobility of device arrays made from MoS₂ films with different stacking sequences and interlayer twist angles. We can see that 30° twisted structure have higher mobility than 0° twisted structure, as electron mobilities of 30°/(30°,0°) twisted samples are higher than that of 0°/(0°, 0°) and (0°, 30°). We attribute the electron mobility of 30° bilayer higher than that of (30°,0°) and (0°, 30°) trilayer to the screen of electric field by the bottom MoS₂ layers, and the enhanced scattering effect by trapped interlayer bubbles in trilayer samples.”

Comments #8: The authors do not discuss possible water contamination that could give rise to the well-known problem of bubbles trapped between the layers.

Author response: Thanks for your comment. To demonstrate the clean surface and interface, we scanned all the surfaces involved in the transfer process. Images are shown as below. Fig. **a-f** are AFM images of **(a)** PDMS surface; **(b)** as grown MoS₂/sapphire surface; **(c)** MoS₂/sapphire surface after stamping and peeling off by PDMS in air; **(d)** MoS₂/PDMS surface; **(e)** surface of transferred monolayer MoS₂ on Si substrate with 300nm SiO₂; **(f)** surface of transferred 30° bilayer MoS₂ on Si substrate with 300nm SiO₂. From Fig. **a**, we can see the surface of PDMS is flat and clean, Ra=677.3pm, no contamination observed. Fig. **b** and **c** show that after PDMS stamping and peeling off in the air, nothing remains on MoS₂/sapphire sample surface, even previous contaminations on twin boundaries disappear. Thus, PDMS will not induce any contamination to MoS₂ surfaces (even can remove contaminations), it is “inert”. Fig. **d** shows that after peeling off MoS₂ from sapphire by PDMS in DI water, the MoS₂/PDMS surface is clean, which means no contamination will be induced during the water assistant transfer process. As illustrated in Fig. **e**, after annealing, the surface of transferred monolayer MoS₂ film is clean and flat. For the bilayer sample in Fig. **f**, we can see most of the area is flat, only a few small bubbles exist (maximum 10%). Bilayer sample has more bubbles than monolayer is due to that the trapped gas cannot escape as interlayer coupling between two MoS₂ layer is much stronger than that between MoS₂ and polycrystal SiO₂. These AFM images demonstrate that our samples have both clean interfaces and surfaces. We put this Figure as Fig. S4 and add “AFM images in Fig. S4a-d indicate that all the surfaces involved in the transfer process are clean and flat, no contamination was observed. Fig.S4e and f show that the surface of transferred monolayer MoS₂ film is also clean and flat, and bilayer sample only has few bubbles (maximum 10% of the area), indicating the high quality of our samples.” to 115 line page 5.

Comments #9: The statement: “Fig. 1a and Fig. S1 are optical images of the as-grown

monolayer MoS₂ film on a 2-inch sapphire wafer, showing that the film is highly uniform and covers the entire wafer.” Seems incorrect to me given that it is not possible to see the uniformity of the film. I recommend to erase this confusing sentence.

Author response: Thanks for your advice, we revised this sentence to “*Fig. 1a illustrates a typical monolayer MoS₂ film on a 2-inch sapphire wafer, the optical microscope images of it in Fig. S1 shows that the film has high uniformity and coverage.*”

Comments #10: Same with the sentence “The optical images indicate that the surfaces of all transferred films are clean and uniform.” This is misleading given that most of the contamination for 2D materials is not visible in optical microscopes.

Author response: Thanks for your suggestion, please refer to AFM images of comment #8.

Comments #11: Scales of figure 2d are difficult to read

Author response: We are sorry for that. The origin word file had enough resolution, but after automatically transformed by the website to PDF the resolution was deteriorated. The scale bar is from 735nm-785nm. In the resubmission, we submitted the figures separately to keep their resolution.

Reviewer #3

The manuscript describes the sequential transfer of grown MoS₂ monolayers into angle controlled bi- or tri-layer stacks. The authors employ a previously reported growth technique, which produces aligned but not single crystal MoS₂ monolayer (there are in fact a lot of domain boundary defects, as mentioned in their prior paper but not here). There have been previous reports on aligned stacking of exfoliated flakes, here the authors do this with the grown mono-layer.

Thanks for your comments. Twist angle between adjacent layers of 2D layered materials provides an exotic degree of freedom to enable various fascinating new phenomena in van der Waals homo- and hetero-structures. We know there are excellent previous studies on the stacking of exfoliated (or grown) flakes, but the processes are quite complex and hard to get clean and large-scale samples. To realize the practical applications of twistrionics, it is of the utmost importance to control the interlayer twist angle on large scales. The main purpose of this article is to offer a new approach to easily fabricate large area, high quality, and relatively precise twist angle controlled multilayer MoS₂ films for optical and electrical proposes.

There are only twin boundaries in our samples, and they are only a small part of the sample area (less than 5%). Our pervious works show that the twin boundaries will not strongly affect the spectral and electrical properties (ACS Nano 11, 12001 (2017); Crystals 2016, 6(9), 115). So, our results should be reliable.

Comments #1: A major claim is the “precisely” or “accurately” controlled angle, which I do not at all find substantiated. Too little detail is given about the transfer methodology and set-up. It is not clear what max. area of transfer can be achieved. Looking at Fig. 2a, unlike claimed, the edges of each layer are not sharp, and the authors must present a statistical discussion and error in alignment. No error bars at all are given, neither is total error of alignment. This is not the standard one would expect to see in Nature journal. I note the error bar discussion needs also to include each characterization technique, and characterization here needs to cover the claimed larger areas to mm size.

Author response: Thanks for your comment. The discussion of the twist angle error is as below. We want to demonstrate the precision and the reproducibility of our method from two parts. First is the method. The linear guide we used to cut the sapphire has maximum parallel misalignment about $20\mu\text{m}/100\text{mm}$, which will induce a maximum 0.01° error. We also changed the diamond head to a tungsten needle and drew a line on $\text{MoS}_2/\text{sapphire}$ sample surface to further test the precision of the linear guide. Coordinates of points on the line were measured by Nikon measuring microscope MM-200. The consequence is as Fig. **a-b** below. For a 30mm range, the misalignment is below 0.005mm, which means a 0.0095° error, agree well with the precision of the linear guide. The rotational stage we used has the resolution 0.033° ($1/30^\circ$), which can provide a relatively precise twist angle between two stack layers.

Second is to directly measure the twist angle of our bilayer samples by TEM electron diffraction from different regions. We tested three 30° twisted bilayer samples. As shown below, Fig. **a** is the optical image of bilayer MoS_2 on a microgrid; Fig. **b-e** are TEM images and electron diffraction patterns from different regions of three 30° twisted bilayer samples; Fig. **f** is the statistics of twist angle. We transferred all twist angles within 30° (red region), and the blue region is just a copy of the red region to make the chart symmetric. From the statistics, we can see the error is around 0.4° . From some electron diffraction, we can see patterns are not perfectly symmetric. This may be due to the strain-induced during the TEM sample preparation, misalignment between the electron beam and c axis of the sample, and wrinkles, which will cause extra errors.

So we can safely claim the error of our method is below 0.4° . We also revised the Fig. 2 and put this statistic into it. And we add “Fig. 2c is the twist angle distribution of three different 30° stacked bilayer MoS_2 films, measured by TEM electron diffraction from different regions. From the statistics, we can see the error is around 0.4° . Because misalign between the electron beam and the c axis of sample, strain, and wrinkles will increase error, we can safely claim that the error of our method is below 0.4° . Please refer to supplementary note 4 for more details.” in line 124 page 5. Please refer to the main text.

Comments #2: Another claim is that interface of transferred films is “ultraclean”. Again this is not substantiated at all. The very small area moire pattern from TEM analysis is certainly not proving anything in this regard. All analysis (incl. the Raman spectra, no maps? Optical maps clearly not sufficient as even dirty films can look homogeneous as is well known) is done over very small (in comparison to wafer growth and mm cuts) areas, even though claim is of scaled, larger sample capability. No (statistically relevant) evidence is provided. I expect to see analysis over whole at least mm-scale. No comments on possible PDMS residual or any other contamination, that can be expected for “wet” processing. Why should the technique give “ultra-clean” interface over mm areas?

Author response: Thanks for your comment. The optical image has the scale \sim mm and the TEM image has scale \sim nm. What lack between mm and nm is μm scale characterization. To demonstrate the clean surface and interface, we scanned all the surfaces involved in the transfer process. Images are shown as below. Fig. a-f are AFM images of (a) PDMS surface; (b) as grown MoS_2 /sapphire surface; (c) MoS_2 /sapphire surface after stamping and peeling off by PDMS in air; (d) MoS_2 /PDMS surface; (e) surface of transferred monolayer MoS_2 on Si substrate with 300nm SiO_2 ; (f) surface of transferred 30° bilayer MoS_2 on Si substrate with 300nm SiO_2 . From Fig. a, we can see the surface of PDMS is flat and clean, $R_a=677.3\text{pm}$, no contamination observed. Fig. b and c show that after PDMS stamping and peeling off in the air, nothing remains on

MoS₂/sapphire sample surface, even previous contaminations on twin boundaries disappear. Thus, PDMS will not induce any contamination to MoS₂ surfaces (even can remove contaminations), it is “inert”. Fig. **d** shows that after peeling off MoS₂ from sapphire by PDMS in DI water, the MoS₂/PDMS surface is clean, which means no contamination will be induced during the water assistant transfer process. As illustrated in Fig. **e**, after annealing, the surface of transferred monolayer MoS₂ film is clean and flat. For the bilayer sample in Fig. **f**, we can see most of the area is flat, only a few small bubbles exist (maximum 10%). Bilayer sample has more bubbles than monolayer is due to that trapped gas cannot escape as interlayer coupling between two MoS₂ layer is much stronger than that between MoS₂ and polycrystal SiO₂. These AFM images demonstrate that our samples have bot clean interfaces and surfaces. We put this Figure as Fig. S4 and add “AFM images in Fig. S4a-d indicate that all the surfaces involved in the transfer process are clean and flat, no contamination was observed. Fig.S4e and f show that the surface of transferred monolayer MoS₂ film is also clean and flat, and bilayer sample only has few bubbles (maximum 10% of the area), indicating the high quality of our samples.” to 115 line page 5.

Comments #3: The data in Fig. 5 b is illegible. Not sufficient detail on mobility analysis is given (at least in SI). Not clear what the error bars refer to in Fig 5c. What is expected contact resistance?

Author response: Thanks for your suggestions. We repeated the device fabrication and characterization. The result is as below. As shown in inset of Fig. a, we used the transmission line structure, so the effect of contact resistance can be excluded. Fig. a and b are electrical transfer and output curves of a standard 30° bilayer device. Fig. c shows the on/off ratio statistics of 0°/30° bilayer devices. Which shows 30° bilayer device have higher on/off ratio. Fig. d is the mobilities of twisted multilayer MoS₂ films, which shows similar phenomenon with pervious devices. We

attribute the electron mobility of 30° bilayer higher than that of (30°,0°) and (0°, 30°) trilayer to the screen of electric field by the bottom MoS₂ layers, and the enhanced scattering effect by trapped interlayer bubbles in trilayer samples. Fig. 5 has been replaced, and we revised to line 226-240 page 10 to “*Inset of Fig. 5a shows the optical image of our device array made from 30° bilayer sample. Devices exhibit high quality and integrity. Standard I/V_g curves are shown in Fig. 5a. Fig. 5b is I/V curves of our device, which are linear under different back gate voltages, showing good contact between MoS₂ and metal electrodes. Fig. 5c is the on/off ratio statistics of 0° and 30° stacked bilayer MoS₂ devices, on/off ratio of our 30°/0° stacked bilayer MoS₂ devices device is $\sim 10^8/10^7$. 30° stacked bilayer MoS₂ devices have higher on/off than 0° is due to 30° stacked bilayer MoS₂ devices have higher on-current (Fig. S10). Fig. 5d is the electron mobility of device arrays made from MoS₂ films with different stacking sequences and interlayer twist angles. We can see that 30° twisted structure have higher mobility than 0° twisted structure, as electron mobilities of 30°/(30°,0°) twisted samples are higher than that of 0°/(0°, 0°) and (0°, 30°). We attribute the electron mobility of 30° bilayer higher than that of (30°,0°) and (0°, 30°) trilayer to the screen of electric field by the bottom MoS₂ layers, and the enhanced scattering effect by trapped interlayer bubbles in trilayer samples.*”

The calculation methods of mobility now in Supplementary Note 7. We add “*For more information, please refer to Supplementary Note 7.*” to line 242 page 10.

Comments #4: I expect also discussion about effect of the many domain boundaries in the films used, as for instance such boundaries are known to trap contamination.

Author response: Thanks for your suggestion. In AFM images of comment #2, we can see there are not too many contaminations on twin boundaries ($<0.5\text{nm}$ height, maybe 5% of the total area). Indeed, twin grain boundaries can affect the PL spectra and usually causes a decrease in PL intensity. Since the size of our laser spot ($\sim 5\ \mu\text{m}^2$) is larger than the grain size ($\sim 1\ \mu\text{m}^2$) and twin grain boundaries are uniformly distributed in the sample, the effects of grain boundaries cannot be observed. Therefore, we can obtain a uniform result. For the electrical properties, please refer to our previous work *Crystals* 2016, 6(9), 115. In which we found “At low temperature, the twin grain boundary (GB) can increase the in-plane electrical conductivity of MoS₂ and the transport exhibits variable-range hopping (VRH), while at high temperature, the twin GB impedes the electrical transport of MoS₂ and the transport exhibits nearest-neighbor hopping (NNH).” Below are the main figures of our previous work.

Figure 2. (a) SEM image of a device with four electrodes contacting two coalesced MoS₂ grains, G_L, GB, and G_R represent the left grain, grain boundary, and right grain; (b) AFM image of region shown in (a).

Figure 3. (a,b) Output characteristics at 80 K and 430 K, respectively; (c) temperature dependence of electrical conductivity σ ; and (d) relative conductivity R_σ at different temperatures.

Reviewer #1 (Remarks to the Author):

The authors have revised the manuscript to address the comments I made in my review, and have made sufficient additions and clarifications to the points raised - therefore I recommend publication of this manuscript. I have only two minor comments.

1. The scale bars of insets of Figure 2d, 5a, S6 and S9 are missing.
2. There are some white dots in the AFM image of Figure S4f. They are bubbles or impurities? In this case, the author should be careful to use the "ultra-clean" in the manuscript.

Reviewer #2 (Remarks to the Author):

The authors have responded to most of my comments/questions, as I wrote before the scientific results seem correct and their interpretation sound. I recommend their manuscript for publication in Nature Communications.

Few remarks about the current state of the manuscript:

- There are too many references to figures in the supplementary report, which makes almost impossible to read the paper without having the support directly available. In my opinion the main text should be sufficient to understand the results without need of the supplementary. The later should be use in case of doubts and need of further explanations.

- I consider calling these structures "ultra-clean" misleading, as stated by the authors only the contamination bubbles made up to 10% of the surface.

Reviewer #3 (Remarks to the Author):

I find that key questions have not been addressed in the revision.

The authors describe the error in revised manuscript and response. I find that very misguided. To start with a statistically relevant set is required, not 3 samples (updated Fig. 2). In rebuttal, they comment: "We think that there are too many issues that can induce errors during the transfer process, and it is almost impossible to ensure an accurate twist angle of every sample. For our method, if we repeat a lot, we can always find samples to fit some strict requests of small twist angles." This is very confusing, as this is why not "ideal case " but actual error discussion is required here ! This is not the standard of error discussion for a Nature journal. It appears in reality the error is rather large for the presented method.

Re cleanliness, the authors now show select AFM images with limited field of view. But this does not address the question initially posed as the claim is that of a scalable method, so cleanliness over large areas (>mm) is the tricky part, not select small area (as this has been shown many times before).

I maintain the methodology is not really novel. The growth part has already been demonstrated by the authors and is a standard powder-based approach that has limited scalability. An as the authors admit there are twin alignments/GBs in the films, so these stacks are not single crystals. The transfer method is a straightforward adaptation of previous works with stamping (e.g.10.1073/pnas.1620140114).

My initial comment was that solid data is required not exaggerated claims. This remains the case, and re the central claims the revised version does not support claims with any solid data. This is not the standard that should be expected for submission to a leading journal like Nat. Comm.

Point-by-point reply to referees' comments and list of changes

Reviewer #1

The authors have revised the manuscript to address the comments I made in my review, and have made sufficient additions and clarifications to the points raised - therefore I recommend publication of this manuscript.

We thank the reviewer for the positive assessment of our revision and appreciate your recommendation to publish our manuscript.

Comments #1: The scale bars of insets of Figure 2d, 5a, S6 and S9 are missing.

Author response: We apologize for the lack of scale bars for insets of Fig. 2d, 5a, S6, and S9. We have revised these Figures and added the scale bars. Please refer to our new revised manuscript.

Comments #2: There are some white dots in the AFM image of Figure S4f. They are bubbles or impurities? In this case, the author should be careful to use the “ultra-clean” in the manuscript.

Author response: Thanks for your comment. The white dots in the AFM image of Fig. S4f should be bubbles because there is no contrast in TEM images (which means no particles at the interface). We have revised the “ultra-clean” to “relatively clean” in the revised manuscript.

Reviewer #2

The authors have responded to most of my comments/questions, as I wrote before the scientific results seem correct and their interpretation sound. I recommend their manuscript for publication in Nature Communications.

We thank the referee for the positive assessment of our work and appreciate your recommendation for publication in Nature Communications.

Comments #1: There are too many references to figures in the supplementary report, which makes almost impossible to read the paper without having the support directly available. In my opinion the main text should be sufficient to understand the results without need of the supplementary. The later should be use in case of doubts and need of further explanations.

Author response: Thanks for your good suggestions. To improve the readability of our paper, we have moved the AFM and TEM results in the supplementary report to Fig. 2 in the main manuscript. The Revised Fig. 2 is as below.

Figure 2 | High quality twisted bilayer MoS₂ films. (a) Optical Images of three typical transferred twisted bilayer MoS₂ films on Si substrates with 300nm SiO₂: 6°, 19°, and 30°. (b) AFM images of the transferred monolayer (left) and 30° bilayer (right) MoS₂ films. (c) STEM image and electron diffraction pattern of 30° stacked bilayer MoS₂ film. (d) Twist angle distribution of eight different 30° stacked bilayer MoS₂ film samples, red dash line is the Gaussian fitting. Blue region is just a copy of the green region, to make the chart symmetric. (e) PL spectrum of 30° stacked bilayer MoS₂ film. Left inset in (e) is the laser scanning confocal fluorescence microscopy image, and the right inset is a 100µm×100µm² mapping of the indirect bandgap position. Source data are provided as a Source Data file.

Comments #2: I consider calling these structures "ultra-clean" misleading, as stated by the authors only the contamination bubbles made up to 10% of the surface.

Author response: We thank the reviewer for the comment. We have revised the "ultra-clean" to "relatively clean" in the revised manuscript.

Comments #1: I find that key questions have not been addressed in the revision. The authors describe the error in revised manuscript and response. I find that very misguided. To start with a statistically relevant set is required, not 3 samples (updated Fig. 2). In rebuttal, they comment: “We think that there are too many issues that can induce errors during the transfer process, and it is almost impossible to ensure an accurate twist angle of every sample. For our method, if we repeat a lot, we can always find samples to fit some strict requests of small twist angles.” This is very confusing, as this is why not “ideal case” but actual error discussion is required here! This is not the standard of error discussion for a Nature journal. It appears in reality the error is rather large for the presented method.

Author response: We accept this comment and apologize for the incomplete response in the last reply. We agree that more statistical data and error discussions are needed.

To meet the standard of error discussion for the journal of Nature Communications, we here present a statistical discussion of error in twist angle. During the last month, we have prepared five new samples with a designed film twist angle of 30° . Then we have eight samples in total, including three old samples. Both Samples are bilayer stacked film on a silicon substrate with a size of $\sim 0.5 \times 0.5 \text{ cm}^2$. Samples were then transferred to TEM grids (3mm diameter copper micro-grids with carbon films). For each TEM sample, we randomly selected 10-20 positions at different grid holes (spacing more than $50 \mu\text{m}$) to measure the twist angle through the electron diffraction. So we have >100 diffraction patterns in total, and film twist angles result are shown in the figures below.

From the original data (left) and statistic data (right), you can see that all eight samples are of good accuracy to the designed film twist angle. The red dash curve is the Gaussian fitting, and the Standard Deviation of the twist angle is $\sigma = 0.327^\circ$. Note that monolayer MoS_2 has threefold rotation symmetry, stacked bilayers with twist angles of $(30+x)^\circ$ and $(30-x)^\circ$ are equivalent in electron diffraction patterns. Therefore, we normalized the measured twist angles all less than 30° (the green region). Blue region is a copy of the green region for Gaussian fitting. Although the accuracy of our method is not as high as that of dry transfer technology ($\sigma \sim 0.1^\circ$) for mechanically exfoliated samples. For the CVD-grown monolayer TMD samples, no previous work achieves to precisely control the interlayer twist angle of centimeter-scale stacked multilayer MoS_2 homostructures.

In the revised manuscript, we have updated the figures (Fig. 2 & Fig.S6) and provided more discussions on page 5:

“...We also fabricated eight 30°-stacked samples and transferred them on TEM grids for further analysis. For each sample, we randomly select 10-20 positions to measure the twist angle through the electron diffraction. The statistics of the measured twist angle is shown as green bars of Fig. 2d (blue bars are mirror copy of green bars for Gaussian fitting). Based on the statistics, the distribution of twist angle is relatively narrow, and the Standard Deviation of the twist angle of our samples is $\sigma = 0.327^\circ$. It is worth noting that, the formation of flat bands in transition metal dichalcogenide (TMD) homo- and hetero- twist structures is not so sensitive to twist angle (spanning over 1°)^{33, 34}. As a consequence, the accuracy of our method is enough to study TMD homo- and hetero-structures based twistrionics. For more detail discussion of the twist angle distribution, please refer to supplementary note 4.”

Figure 2 | High quality twisted bilayer MoS₂ films. (a) Optical Images of three typical transferred twisted bilayer MoS₂ films on Si substrates with 300nm SiO₂: 6°, 19°, and 30°. (b) AFM images of the transferred monolayer (left) and 30° bilayer (right) MoS₂ films. (c) STEM image and electron diffraction pattern of 30° stacked bilayer MoS₂ film. (d) Twist angle distribution of eight different 30° stacked bilayer MoS₂ film samples, red dash line is the Gaussian fitting. Blue region is just a copy of the green region, to make the chart symmetric. (e) PL spectrum of 30° stacked bilayer MoS₂ film. Left inset in (e) is the laser scanning confocal fluorescence microscopy image, and the right inset is a 100µm×100µm² mapping of the indirect bandgap position. Source data are provided as a Source Data file.

Figure S6 | Electron diffraction characterization of different regions on three 30° twisted bilayer samples. (a) Optical image of bilayer MoS₂ film on a TEM grid. (b)-(e) TEM images and electron diffraction patterns from different regions of three 30° twisted bilayer samples. (f) Twist angle distribution of different samples. Source data are provided as a Source Data file.

Comments #2: Re cleanliness, the authors now show select AFM images with limited field of view. But this does not address the question initially posed as the claim is that of a scalable method, so cleanliness over large areas (>mm) is the tricky part, not select small area (as this has been shown many times before).

Author response: The mm-scale evaluation of cleanliness can be found in the optical microscope image of Figure 2a. Besides, the mapping of the indirect bandgap position (inset of Fig. 2e and Fig. S7) over large areas (200×200 μm²) is also very uniform. These optical images show a uniform and clean stacked structures but lack of micro-scale information, thus we performed AFM. Due to the limited scanning area of AFM, we have to select a small area for imaging. But please note that these imaging areas were randomly selected here and there (like the four typical images shown below, from four different samples) in the mm-scale stacked films, these images should be real and present the overall cleanliness of a film. To be more rigorous, we have revised the “ultra-clean” to “relatively clean” in the revised manuscript.

As for transferring CVD grown samples, bubbles and wrinkles are inevitable due to the strain during the “peeling” and “stacking” process. Although there are some local bubbles and folds (up to 10% of the surface), the overall uniformity is good. From the TEM images in Fig. S6, we didn’t

observe any contrast at the interface, suggesting those brighter points in AFM image are bubbles but not particles. Besides, supported by Fig. S4, our approach does not involve chemicals that will seriously contaminate the surfaces.

Comments #3: I maintain the methodology is not really novel. The growth part has already been demonstrated by the authors and is a standard powder-based approach that has limited scalability. As the authors admit there are twin alignments/GBs in the films, so these stacks are not single crystals. The transfer method is a straightforward adaptation of previous works with stamping (e.g.10.1073/pnas.1620140114).

Author response: Thanks for your detailed comments. “Dry” transfer techniques with highly accurate rotational alignment have been developed (10.1021/acs.nanolett.5b05263 and 10.1073/pnas.1620140114, cited as Ref. 17 and Ref. 18). Such ‘tear and stack’ techniques are clean and suitable for high-quality stacked structures at a micro-scale. But larger-scale twist angle control stacking remains a challenge. We agree that both the transfer process and the growth part are not really novel and have reported previously. However, transfer and stacking monolayer TMD at a large scale with twist angle control has not been demonstrated yet. From this point of view, we believe it is novel. Our work would offer a possible route in controlling the twist angle of 2D material at a wafer-scale.

The referee is right. Our monolayer MoS₂ films have twin alignments and GBs. The stacked samples are not single crystals. If we design the film twist angle to be α ($\alpha \leq 30^\circ$), then the stacked bilayer MoS₂ should have two lattice twist angles of α and $60^\circ - \alpha$. According to previous studies (like Ref.31: Nature Communications 5, 4966 (2014)), both the interlayer coupling and interlayer distance of these two structures are almost identical. The optical and electrical properties between these two structures are nearly the same. For example, WSe₂/WS₂ moiré superlattice with twist angle 0° and 60° show almost the same triangular lattice Hubbard physics (Ref. 32: arXiv preprint arXiv:1910.08673). Thus we conclude that the present twist-angle controlled stacking of monolayer TMDs with twin alignments/GBs is still meaningful. To avoid misunderstanding, we have provided more discussions in the revised manuscript on page 5:

“...Notice that, due to the threefold rotation symmetry of monolayer MoS₂ lattice and the existence of twin lattice alignments in these films, the bilayer samples with transfer stack angles θ should have both θ and $60^\circ - \theta$ lattice twist angles regions, which have the same electron diffraction patterns. According to previous studies, both the interlayer coupling and interlayer distance of these two structures are almost identical³¹; the optical and electrical properties between these two structures are also similar (For example, WSe₂/WS₂ moiré superlattice with twist angle 0° and 60° show nearly the same triangular lattice Hubbard physics³²). Thus, the properties of the transferred multilayer MoS₂ films can be controlled by the stacking angle of as-grown monolayers with twin alignments. In this paper, we directly marked all the measured twist angle within 30° for simplicity.”

Comments #4: My initial comment was that solid data is required not exaggerated claims. This remains the case, and re the central claims the revised version does not support claims with any solid data.

Author response: Thanks for your time and work to review this paper. According to your comments and suggestions, we have carried out more experiments and included new data and discussions in the revised manuscript. We indeed have tried our best to make this work as solid as possible and avoid exaggerated claims. Hopefully, the revised version can fully address your comments and meet with your criteria.

Reviewers' comments:

Reviewer #2 (Remarks to the Author):

The authors addressed my comments and I have no further questions. I recommend this article for publication.

Reviewer #3 (Remarks to the Author):

I appreciate that the authors did more samples, now 8, and analysed them in detail. I am fully aware of how much work that is. It definitely improved the paper. I note though that the standard deviation figure included in the main manuscript (2d), now has mixed up sample-to-sample and within one sample variation, and I would prefer to see this separated. Also this deviation is for 30deg, the authors should comment if they expect deviation to be the same for other angles. Also this includes possible issues due to films not really being single crystal.

I would encourage the authors to clearly state if in STEM/SAED analysis any filters were used during the data recording and processing. The STEM images appear to be enhanced. If we compare the FFT of HAADF-STEM image (left) and the SAED image (right), we notice two things. A.) There are haloes around the FFT 'diffraction spots'. This could be due to image compression, but more likely due to an image filter applied. B.) There are extra spots in the FFT of the HAADF-STEM image but not in the SAED. One of them is circled out. The authors should comment on that.

Also some other details should be added to ensure reproducibility: transfer: "were annealed at 400 °C for 8 hours, under the protection of H₂/Ar gas", here I expect to see total pressure, and gas ratio. Devices: "then the standard UV-lithography process was", here I expect to see resists/developer used.

The abstract still reads that precise control over nm-scale is possible, which implies single crystal and clean samples. This should still be clarified.

The authors now include larger range AFM images that show a more realistic picture. Why did the authors not include all this data in the manuscript or SI? The authors are as limited in cleanliness as everybody else in the field, so I find it completely unnecessary and unscientific to use adjectives like "ultra-clean". The authors now write in one case "relatively clean", but throughout the manuscript still claim the interface, samples etc are clean. This is still a bit misleading, as it just refers to limits of characterisation. This should still be clarified.

I appreciate that the authors did more samples, now 8, and analysed them in detail. I am fully aware how much work that is. It definitely improved the paper. I note though that the standard deviation figure included in main manuscript (2d), now has mixed up sample-to-sample and within one sample variation, and I would prefer to see this separated. Also this deviation is for 30deg, the authors should comment if they expect deviation to be the same for other angles. Also this includes possible issues due to films not really being single crystal.

I would encourage the authors to clearly state if in STEM/SAED analysis any filters were used during the data recording and processing. The STEM images appear to be enhanced. If we compare the FFT of HAADF-STEM image (left) and the SAED image (right), we notice two things. A.) There are haloes around the FFT 'diffraction spots'. This could be due to image compression, but more likely due to image filter applied. B.) There are extra spots in the FFT of the HAADF-STEM image but not in the SAED. One of them is circled out. The authors should comment on that.

Also some other details should be added to ensure reproducibility: transfer: "were annealed at 400 °C for 8 hours, under the protection of H₂/Ar gas", here I expect to see total pressure, and gas ratio. Devices: "then the standard UV-lithography process was", here I expect to see resists/developer used

The abstract still reads that precise control over cm-scale is possible, which implies single crystal and clean samples. This should still be clarified.

The authors now include larger range AFM images that shows more realistic picture. Why did the authors not include all this data in the manuscript or SI? The authors are as limited in cleanliness as everybody else in field, so I find it completely unnecessary and unscientific to use adjectives like "ultra-clean". The authors now write in one case "relatively clean", but throughout the manuscript still claim the interface, samples etc are clean. This is still bit misleading, as it just refers to limits of characterisation. This should still be clarified.

Point-by-point reply to referees' comments and list of changes

Reviewer #3

Comments #1: I appreciate that the authors did more samples, now 8, and analyzed them in detail. I am fully aware how much work that is. It definitely improved the paper. I note though that the standard deviation figure included in main manuscript (2d), now has mixed up sample-to-sample and within one sample variation, and I would prefer to see this separated. Also this deviation is for 30deg, the authors should comment if they expect deviation to be the same for other angles. Also this includes possible issues due to films not really being single crystal.

Author response: Thank you for your comment. In Fig. S6f, we show the raw data of different samples in Fig. 2d with different colors. Please refer to the revised supporting information for more details. The Fig. S6 and Std Dev of samples 1-8 are as below:

Figure S6 | Electron diffraction characterization of different regions on three 30° twisted bilayer samples. (a) Optical image of bilayer MoS₂ film on a TEM grid. **(b)-(e)** TEM images and electron diffraction patterns from different regions of three 30° twisted bilayer samples. **(f)** Twist angle distribution of different samples. Source data are provided as a Source Data file.

Std Dev of samples 1-8							
sample 1	sample 2	sample 3	sample 4	sample 5	sample 6	sample 7	sample 8
0.170018°	0.23516°	0.085391°	0.12106°	0.169337°	0.253908°	0.09912°	0.271182°

We think that the twist angle deviation of all stack samples is similar because the preparation process of them are identical. On twin boundary issue, we have included the following discussions in the newly revised manuscript: “Notice that, due to the threefold rotation symmetry of monolayer MoS₂ lattice and the existence of twin lattice alignments in these films, the bilayer samples with transfer stack angles θ should have both θ and $60^\circ - \theta$ lattice twist angles regions, which have the same electron diffraction patterns. According to previous studies, both the interlayer

coupling and interlayer distance of these two structures are almost identical³¹; the optical and electrical properties between these two structures are also similar (For example, WSe_2/WS_2 moiré superlattice with twist angle 0° and 60° show nearly the same triangular lattice Hubbard physics³²). Thus, the properties of the transferred multilayer MoS_2 films can be controlled by the stacking angle of as-grown monolayers with twin alignments. In this paper, we directly marked all the measured twist angle within 30° for simplicity.”

Comments #2: I would encourage the authors to clearly state if in STEM/SAED analysis any filters were used during the data recording and processing. The STEM images appear to be enhanced. If we compare the FFT of HAADF-STEM image (left) and the SAED image (right), we notice two things. A.) There are haloes around the FFT ‘diffraction spots’. This could be due to image compression, but more likely due to image filter applied. B.) There are extra spots in the FFT of the HAADF-STEM image but not in the SAED. One of them is circled out. The authors should comment on that.

Author response: Sorry, we didn’t specify it. We indeed applied FFT filtering to the image to enhance the sharpness. Figures below show our FFT filtering process. Fig. **a** is the raw STEM image and Fig. **b** is the FFT pattern of Fig. **a**. In Fig. **c** we selected the main points of Fig. **b** as mask regions and made a reverse FFT to it, then we get the final enhanced image as Fig. **d**. In Fig. **b** there is no haloes or extra spots, so haloes or extra spots origin from the FFT filtering process. For the SAED, we didn’t apply any filter to it.

In the revised manuscript, we included the description of Fig. 2c as “*STEM image after FFT filtering and electron diffraction pattern of 30° stacked bilayer MoS_2 film.*”

Comments #3: Also some other details should be added to ensure reproducibility: transfer: “were annealed at 400 °C for 8 hours, under the protection of H₂/Ar gas”, here I expect to see total pressure, and gas ratio. Devices: “then the standard UV-lithography process was”, here I expect to see resists/developer used.

Author response: Thanks for your suggestions. The anneal gas ratio is Ar 150 sccm/ H₂ 20 sccm, the total pressure is ~ 1 Torr. We used AR-P 5350 photoresist and AR 300-47 developer. We have added these details into the revised method:

“The transferred twisted multilayer MoS₂ films were firstly patterned with RIE (Plasma Lab 80 Plus, Oxford Instruments Company) by oxygen plasma, and then the standard UV-lithography (MA6, Karl Suss) process was used to pattern source/drain contacts with AR-5350 as the photoresist, which was spin coated on sample surface at 4000 rpm and baked at 100 °C for 4 min. The developer is AR 300-47.”

Comments #4: The abstract still reads that precise control over cm-scale is possible, which implies single crystal and clean samples. This should still be clarified.

Author response: Thank you for your comment. We think that by our method it’s possible achieved precise control of twist angle on cm-scale multilayer MoS₂. The influence of the twin boundary has been discussed in the revised manuscript and comment #1. The interface cleanness has been discussed in S3 in supporting information as below:

Figure S4 | AFM images of all surfaces involved in transfer processes. (a) PDMS surface. **(b)** As grown MoS₂/sapphire surface. **(c)** MoS₂/sapphire surface after stamping and peeling off by PDMS in air. **(d)** MoS₂/PDMS surface. **(e)** Surface of transferred monolayer MoS₂ on Si substrate with 300nm SiO₂. **(f)** Surface of transferred 30° bilayer MoS₂ on Si substrate with 300nm SiO₂. Source data are provided as a Source Data file.

“Fig.S4 show the AFM images of all surfaces involved during the transfer process. From Fig. S4a, we can see the surface of PDMS is flat and clean, Ra=677.3pm, no contamination was observed. After PDMS stamping and peeling off in the air, nothing remains on MoS₂/sapphire sample surface, even previous

contaminations on twin boundaries disappear (Fig. S4b and c). Thus, PDMS will not induce any contaminations to MoS₂ surfaces (even can remove contaminations), it is “inert”. Fig. S4d shows that after peeling off MoS₂ from sapphire by PDMS in DI water, the MoS₂/PDMS surface is still clean, which means no contamination will be induced by DI water. As illustrated in Fig. S4e, after annealing, the surface of transferred monolayer MoS₂ film is clean and flat. For the bilayer sample in Fig. S4f, most of the area is flat, only a few small bubbles exist (maximum 10%). Bilayer sample has more bubbles than monolayer is due to interlayer coupling between two MoS₂ layer is much stronger than that between MoS₂ and polycrystal SiO₂, which trapped gas cannot escape. These AFM images demonstrate our samples have both clean interfaces and surfaces.”

We didn't observe any contamination on all the surfaces involved in the transfer process.

Comments #5: The authors now include larger range AFM images that shows more realistic picture. Why did the authors not include all this data in the manuscript or SI? The authors are as limited in cleanness as everybody else in field, so I find it completely unnecessary and unscientific to use adjectives like “ultra-clean”. The authors now write in one case “relatively clean”, but throughout the manuscript still claim the interface, samples etc are clean. This is still bit misleading, as it just refers to limits of characterization. This should still be clarified.

Author response: Thank you for your comment. The reason we didn't put all AFM topography images of bilayer sample is that we think AFM topography images of bilayers are not sufficient to show interface cleanness of them. This is because AFM can give the information of samples' surfaces, but it can't directly scan interfaces. So, even the AFM topography image is flat, it doesn't mean the interface is clean.

Instead, we gave AFM images of all the surfaces involved in the transfer process (as-grown MoS₂ sample, PDMS, MoS₂ on PDMS, MoS₂ surface after PDMS stamp and peeling, transferred monolayer and bilayer samples, please also refer to comment #4) in Fig. S4 to show the interface cleanness of our sample. These AFM images mean that: first, the surfaces of MoS₂ films are always clean at every step of transferring; second, PDMS and DI water will not cause any residue on MoS₂ surfaces. So, the interfaces of transferred MoS₂ multilayer films should be clean. The bright points and wrinkles on the AFM image of transferred bilayer MoS₂ films are just bubbles. So, compared with the common large scale “wet” transfer process (which needs etching and dissolve the transfer polymer medium), our transfer method minimizes the interface contamination for large scale samples. In this case, we think that our samples are “relatively clean”.

REVIEWERS' COMMENTS:

"I have carefully read the response from the author. The author have addressed each individual point in detail. I believe the paper is sufficient for publication.

One minor suggestion about the comments 3, the author should add the detailed parameters (ie. temperature, pressure, gas) of anneal treatment in the manuscript rather than just response to the reviewer, since the vacuum anneal treatment is important to achieve the clean surface. This will offer valuable information for some readers who want to repeat the experiment or do the transfer by the same method."

Point-by-point reply to referees' comments and list of changes

Comments #1: I have carefully read the response from the author. The author have addressed each individual point in detail. I believe the paper is sufficient for publication.

One minor suggestion about the comments 3, the author should add the detailed parameters (ie. temperature, pressure, gas) of anneal treatment in the manuscript rather than just response to the reviewer, since the vacuum anneal treatment is important to achieve the clean surface. This will offer valuable information for some readers who want to repeat the experiment or do the transfer by the same method.

Author response: Thank you for your suggestion. We already added the detailed parameters of anneal treatment in the Method.